# Utilizing river and wastewater as a SARS-CoV-2 surveillance tool in settings with limited formal sewage systems

Kayla G. Barnes [1,2,3,4] ✉, Joshua I. Levy [4], Jillian Gauld[5], Jonathan Rigby [1,6], Oscar Kanjerwa[1], Christopher B. Uzzell [7], Chisomo Chilupsya[1], Catherine Anscombe[1,6], Christopher Tomkins-Tinch [2,8], Omar Mbeti[9], Edward Cairns[10], Herbert Thole[1], Shannon McSweeney[1,6], Marah G. Chibwana [1], Philip M. Ashton[1,10], Khuzwayo C. Jere [1,10,11], John Scott Meschke[12], Peter Diggle[13], Jennifer Cornick[1,10], Benjamin Chilima[14], Kondwani Jambo[1,5,15], Kristian G. Andersen [4,16], Gift Kawalazira[9], Steve Paterson [10], Tonney S. Nyirenda[1,17] & Nicholas Feasey [1,6,18]

The COVID-19 pandemic has profoundly impacted health systems globally and robust surveillance has been critical for pandemic control, however not all countries can currently sustain community pathogen surveillance programs. Wastewater surveillance has proven valuable in high-income settings, but less is known about the utility of water surveillance of pathogens in low-income countries. Here we show how wastewater surveillance of SAR-CoV-2 can be used to identify temporal changes and help determine circulating variants quickly. In Malawi, a country with limited community-based COVID-19 testing capacity, we explore the utility of rivers and wastewater for SARS-CoV-2 surveillance. From May 2020–May 2022, we collect water from up to 112 river or defunct wastewater treatment plant sites, detecting SARS-CoV-2 in 8.3% of samples. Peak SARS-CoV-2 detection in water samples predate peaks in clinical cases. Sequencing of water samples identified the Beta, Delta, and Omicron variants, with Delta and Omicron detected well in advance of detection in patients. Our work highlights how wastewater can be used to detect emerging waves, identify variants of concern, and provide an early warning system in settings with no formal sewage systems.

As the COVID-19 pandemic continues, it has become apparent that globally representative tracking of disease trends and rapid identification of novel variants is essential to pandemic control[1]. Whilst high-income countries have high rates of vaccine coverage, including booster campaigns, low-income countries (LICs) lag in comparison, creating environments where severe acute respiratory syndrome coronavirus 2 (SARS-CoV-2) can continue to circulate and mutate, especially in settings with high prevalence of immunosuppressive illness[2,3]. In Malawi, there is limited capacity within the health care system to

carry out large-scale surveillance of COVID-19 in the community, especially between waves, which limits the ability of public health services. An early warning system to detect increasing cases and identify imported or novel variants of concern (VOCs) is therefore urgently required. We therefore aimed to first establish SARS-CoV-2 detection in wastewater and then model if an ES program can predict peaks in SARS-CoV-2 and identify genomic variants of concern.

Genomic surveillance of SARS-CoV-2 is currently limited to two Malawian cities (Blantyre and Lilongwe) and is further limited due to

low sequencing resources and capacity. In Malawi, sero-surveys have shown high levels of exposure consistent with observations in high income countries, with 27.55% of healthcare workers testing positive for anti-SARS-CoV-2 antibodies as early as 2020[4]. Further, in 2021 over 80% of local blood bank samples were positive for antibodies[5]. There have also been high levels of asymptomatic infection (45.7%)[6]. These studies show vastly higher circulation of SARS-CoV-2 than reported through surveillance of individuals presenting with clinical features of COVID-19 to secondary or primary care facilities. Although there is high exposure in Malawi providing some population-level immunity by the end of 2022 only around 13% of the population was vaccinated[7] against COVID-19, leading to a population at greater risk for circulating VOCs. The disparity between genomic surveillance and clinical case-loads has created a need for novel surveillance strategies that can survey a larger proportion of the population at low enough cost to be sustainable, yet which offer the resolution to predict trends in clinical disease and identify emerging VOCs.

The utility of wastewater surveillance to identify VOCs and track COVID-19 has been well established[8–12] but most countries lack wastewater treatment plants and closed sewage systems. SARS-CoV-2 has been identified in rivers in Japan[13] and Nepal[14], open sewage in Peru[15], Nepal[14] and surface water in low-income communities in Buenos Aires[16] but overall there have been few examples of the utility of wastewater surveillance outside high-income settings. SARS-CoV-2 concentrations in feces are typically in the range of 10^3 to 10^7 RNA copies per gram, in more than 50% of symptomatic patients[17], and further fecal viral shedding can continue even after symptoms subside[18] therefore wastewater sources are an important tool to track SARS-CoV-2. SARS-CoV-2 RNA shed into wastewaters can be detected, quantified, and sequenced from wastewater to act as a proxy for transmission, estimate prevalence, and track variants[19–24]. Finally, recent advances in computational analysis of wastewater samples allows for both clinically observed lineage and cryptic lineage detection, i.e., virus lineages that are not being detected in the human population but are still circulating[25].

Globally, only 52% of sewage and wastewater is treated and this drops to only 4.2% in low-income countries (LICs)[26]. In Blantyre, the 2nd largest city in Malawi there are no active sewage treatment plants. Instead, the population largely uses earthen (67%) or concrete (20%) latrines which often lead to fecal contamination of groundwater and river systems. Sanitation infrastructure deficits are common to many LICs and it is unclear whether and how surveillance of the contaminated environment (henceforth environmental surveillance or ES) can be used as a public health tool to track SARS-CoV-2 and inform decision and policy makers[19,27,28]. Our recent work highlighted how ES can be a cost-effective tool in LICs[29] and here we describe how river and wastewater surveillance can be used for monitoring trends in SARS-CoV-2 compared to clinical surveillance and identifying variants of concern.

## Results

### SARS-CoV-2 detection in river and sewage

Water from rivers which act as informal sewage lines and the defunct Manase wastewater treatment plant (WWTP) was collected in two phases. Phase 1, from May 2020–January 2021, we developed our sample framework, field sampling method and laboratory methods and showed consistent detection of SARS-Cov-2 using seven collection sites. During Phase 2, January 2021–May 2022, we expanded our collection sites to cover the entire city of Blantyre, Malawi (Fig. 1B) and capture close to 100% of the population. The city of Blantyre is surrounded by 3 small mountains (Fig S1a) and the defunct Manase WWTP represents the bottom of the city with only 2 river sites downstream. Collection sites are based on previous work[30–32] (described in the methods) and almost all 'rivers' have fecal contaminate (Rigby, Feasey et al. *in prep*). In addition to SARS-CoV-2 we tested for Pepper Mild

Mottle Virus (PMMoV)[33] but after testing numerous wastewater samples and stool samples we never found the presence of PMMoV. In Malawi there are only a few varietals of peppers, and we believe PMMoV is not present in high concentration in Malawi. We also tested a subset of matched water samples for the 16S rRNA gene of *Bacteroides dorei* Hf183[34] as part of our matched S. Typhi detection work (*in prep*) and found around 55% (range by site 14–85%) of samples were positive for Hf183.

Phase 1 was a proof of principle study that included collections in 7 sites spanning 3 urban areas and the defunct WWTP. These sites were chosen based on previous work showing the presence of S. Typhi, so we had prior evidence of fecal contamination and pathogen detection[31,32]. We first detected SARS-CoV-2 on May 11th, 2020, and found presence of the virus in all sites tested with the defunct WWTP having the most positive samples (Supplementary Data 2). The defunct WWTP includes run off from multiple large rivers including the largest water system in Blantyre (Mudi River) and is one of the lowest points of the city. In addition, the WWTP is mainly stagnant water where virus and other organic materials accumulate.

In Phase 2 we scaled the collection to 112 sites, which included the original seven sites, covering 22 areas representing the entire river system in Blantyre and therefore most of the population (Fig. 1B, Fig S1c, Supplementary Data 1, 2). The scaled-up program had two main objectives: (1) to determine where we could detect SARS-CoV-2 and (2) to comprehensively sample the entire population of Blantyre to provide a comparator to clinical surveillance.

Overall, from May 2020-May 2022 we found 8.3% (220/2625) of our samples were positive for SARS-CoV-2 by PCR, with Ct range of 30–39.45 (Fig S2, Supplementary Data 1). We found 70/112 sites did not have any detectable SARS-CoV-2. For many sites we were only able to collect samples <3 times in the 2-year period. This was due to a three main factors: changing river ways during and after raining season, inaccessibility to return to sites due to degradation of roads, and misinformation about our study (in the case of the Bangwe sites) that was corrected with community engagement. Sampling rates and positive detections varied over time (Fig. 1A) and by site (Fig. 1C, Fig S1c). Samples from the defunct WWTP had higher SARS-CoV-2 positivity rates (21% [55/258]) than river water samples (7.2% [170/2367]) (Supplementary Data 1), however river water samples still yielded a signal which varied in line with the occurrence of new waves of COVID-19 in Blantyre. One or more sample(s) had detectable SARS-CoV-2 at 42/112 sites (Fig. 1B, C). Using Anselin Local Moran's $I$ and Getis-Ord $G_i^*$ analysis we identified statistically significant high-high clusters and hotspots of positivity, respectively. Specifically, increased rates of detection were geographically centered toward the southwest of Blantyre along the Mudi and Naperi rivers which drain through the most densely populated areas of the city (Fig. 1D, E). Moreover, local Moran's $I$ also identified statistically significant low-low clusters of positivity in the southeast and north of Blantyre indicating areas of significantly low rates of detection. Trends in positivity did not correlate with rainfall which occur typically from December-May and could cause run off and dilution of water sources (Fig S1d).

### Validation of ES as a predictor of clinical prevalence

Next, we aimed to validate ES as a predictor of clinical cases utilizing two clinical surveillance datasets. The clinical dataset we utilized were an active recruitment study and the passive District Health Office (DHO) dataset. For the active surveillance from November 2020 to July 2022 patients were recruited in two of the largest clinics. A maximum of 20 individuals/day who met the WHO syndromic definition of COVID-19 were tested as well as up to 10 non-syndromic patients to capture asymptomatic transmission[35]. This active community surveillance program tracked both symptomatic and asymptomatic cases as well as the total number of diagnostic tests conducted utilizing a

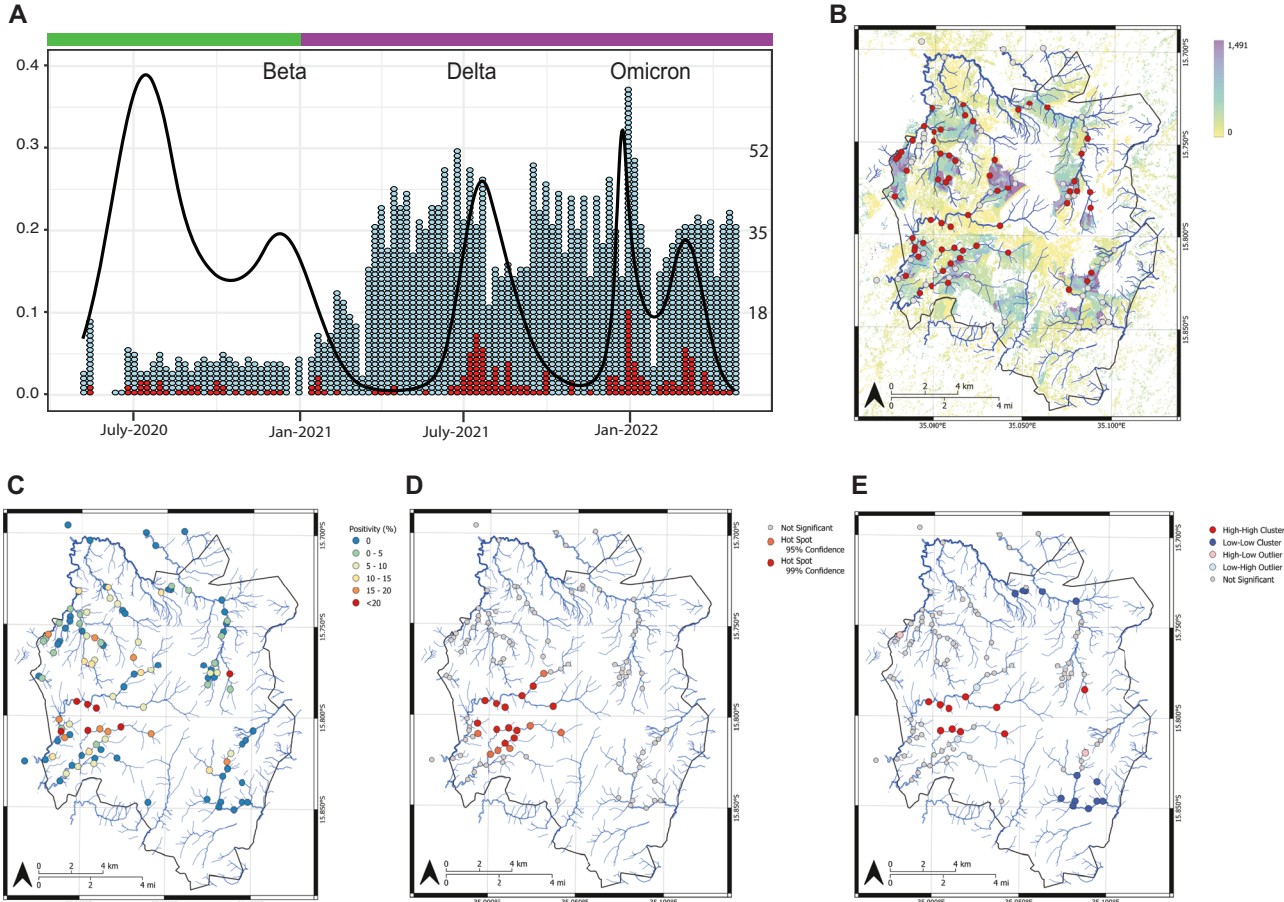

**Fig. 1 | Temporal ES sampling in Blantyre Malawi. A** Sampling over time where each individual dot is one sample tested either negative (blue) or positive (red) for SARS-CoV-2. The left y-axis and black line are the Spline curve modeling peaks and valleys in detection based on the frequenting of positivity. The right y-axis is total of number of samples tested. Phase one is denoted with a green top bar and included 7 sites and phase 2 is denoted with a purple bar and includes up to 112 sites. Utilizing the full dataset from May 2020-May 2022 we analyzed collection sites based on their GPS coordinate. **B** Red dots denote sites with at least one positive sample overlaid on the population density of Blantyre based on HRSL data. **C** each sampling location is color coded by the overall percent positivity. **D** Hotspot analysis using Getis-Ord $G_i^*$ and **E** spatial cluster-outlier detection analysis using Anselin Local Moran's *I*.

tablet-based LIMS system. The passive dataset relied on community clinics reporting positive cases per day to the DHO. This passive dataset contained daily counts of reported city-wide cases and was available from the beginning of ES sampling until January 2022, and represented case detection from symptomatic patients seeking treatment and limited contact tracing. A large proportion of individuals presented to secondary community health centers which is representative of data more typically available in low-income settings. There were key limitations to this dataset. The DHO dataset did not record the total number of tests administered and there was limited testing of asymptomatic individuals. In addition, there were periods of low testing due to PCR resources and there was a national switch from PCR based diagnostics to rapid diagnostic tests, but this was not captured in the dataset available.

### Comparison of the peaks of SARS-CoV-2 from ES in relation to clinical datasets

We compare peaks of SARS-CoV-2 detections to both the active dataset (Fig. 2) and the passive dataset (Fig S3). Examining all three datasets we identified potential discrepancies in the DHO passive surveillance where the Delta wave appears significantly smaller than the other waves, with the Omicron wave contained 3 days at the end of the year that led to a spike in positivity that we did not observe in other datasets (Fig S3), thus distinct from the ES and community-based active surveillance data (Fig. 2).

We compared the timing of the peak of SARS-CoV-2 detection across all three datasets. We first estimated the lag between ES and clinical prevalence. The passive (DHO) data set lagged behind ES data by 2, 31, and 7 days for Wave 1, Beta wave and Omicron wave, respectively, however for the Delta wave the ES data lagged behind the passive dataset by 13 days (Table 1). The active dataset lagged behind ES by almost two months for Beta, but this was at the beginning of this study when recruitment and clinical teams were just being established. For Delta and Omicron there is almost no lag which we believe is being driven by both symptomatic and asymptotical testing capturing early cases similar to wastewater surveillance. Despite the differences between active surveillance, passive surveillance and ES, the peaks of the waves were very similar, further validating ES as a useful indicator for increases in COVID-19 in any setting.

### Modeling ES as a predictor of clinical cases

Using a quasi-binomial generalized linear model, we found that the patterns of SARS-CoV-2 detection in ES samples significantly predict the rate of positivity in the active surveillance system for all waves ($p < 0.001$, Wald test) (Table 2). We chose to model the active surveillance system as we were more confident that the results were accurate for cases (both symptomatic and asymptomatic) per day with a known overall test number. The estimated intercepts in the model can inform the sensitivity of the ES system, and we find an ES detection rate of zero corresponds with approximately 5–6% prevalence in the

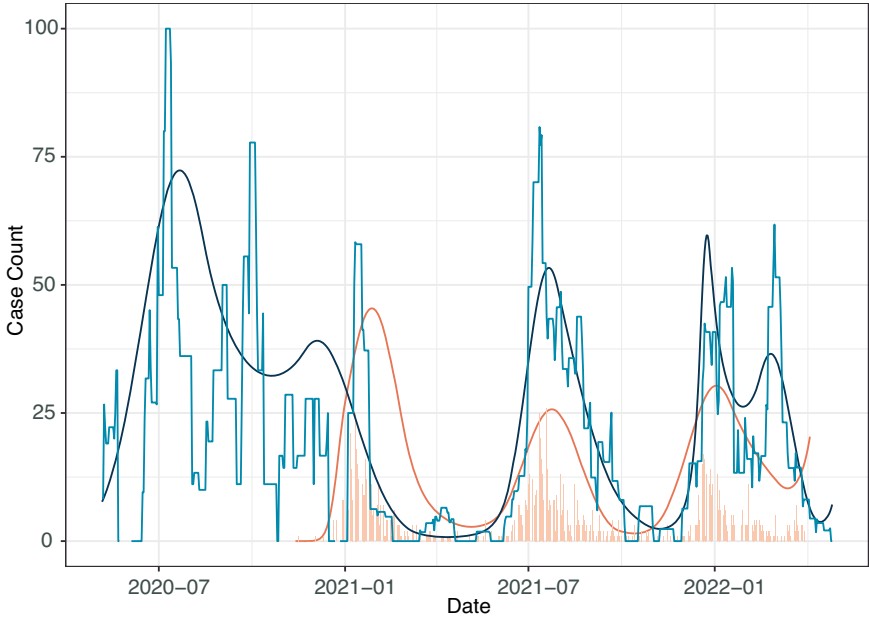

**Fig. 2 | Estimates of lag time between ES and active case surveillance.** Light blue represents the rolling average of ES positivity over time compared to active case surveillance numbers (bar graph in light orange) where the y-axis is total on number of positive cases per day. Spline comparison of both ES (dark blue) and active case surveillance prevalence (dark orange) show how closely linked peaks in both detection methods are over multiple waves.

**Table 1 | Estimated date of peak detection and estimated tolerance interval**

| Wave (VOC) | Environmental Surveillance (95% tolerance interval) | Active Surveillance (95% tolerance interval) | Passive Surveillance (DHO) (95% tolerance interval) |
|---|---|---|---|
| Wave 1 | 7-13-2020 (5-28-2020, 8-01-2020) | NA | 7-15-2020 (7-14-2020, 7-17-2020) |
| Wave 2 (Beta) | 12-05-2020 (10-29-2020, 12-23-2020) | 1-27-2021 (1-22-2021, 1-31-2021) | 1-05-2021 (1-04-2021, 1-06-2021) |
| Wave 3 (Delta) | 7-22-2021 (7-14-2021, 8-04-2021) | 7-22-2021 (7-19-2021, 7-25-2021) | 7-09-2021 (7-08-2021, 7-12-2021) |
| Wave 4 (Omicron) | 12-28-2021 (12-22-2021, 3-07, 2022) | 12-30-2022 (12-26-2021, 1-05-2022) | 1-04-2022 (1-03-2022, 1-05-2022) |

Timing of the maximum detection rates for each wave estimated from environmental surveillance, community active surveillance, and community passive surveillance, respectively. Both active and passive detection lags behind ES detection, except during the Delta wave. Dates indicate the detected peak with 95% tolerance showing the date range of the peaks.

**Table 2 | Estimated model parameters for the predictive model of Covid cases showing wave-specific coefficients for the ES rate as a predictor (α), as well as wave-specific intercepts (β) denoted for each wave driven by a VOC**

|  | Estimate | Std. Error | p-value |
|---|---|---|---|
| $\alpha_{Beta}$ | −4.659 | 0.494 | <0.001 |
| $\alpha_{Delta}$ | −2.908 | 0.168 | <0.001 |
| $\alpha_{Omicron}$ | −2.771 | 0.203 | <0.001 |
| $\beta_{Beta}$ | 23.326 | 2.950 | <0.001 |
| $\beta_{Delta}$ | 10.241 | 0.894 | <0.001 |
| $\beta_{Omicron}$ | 11.405 | 1.197 | <0.001 |

P-value based on the Wald test.

community in the Delta and Omicron waves, indicating lack of sensitivity of the ES system below this prevalence rate (Table 2). As COVID-19 increased in the community, with each wave driven by a new VOC, so did detection of SARS-CoV-2 in wastewater.

Further, we detected differences in wave-specific sensitivities of the ES system indicated by the $\beta$ parameter (Table 2, Fig S6). A 20% detection rate in ES during the beta-wave corresponded to a 50% prevalence rate in the community. The same rate of detection in ES would correspond to a 30% prevalence rate if it occurred in the delta wave, indicating a possible decline in healthcare seeking or reporting in the second wave.

Finally, we tested the robustness of these estimates to uncertainty in the ES data. We generated 1000 realizations from the multivariate normal distribution parameterized by the ES regression model covariates (Fig S4). The realizations were significant ($p < 0.05$) for 100% and 99.9% of iterations for Delta and Omicron, respectively, and 92% for the Beta wave (Fig S7), reflecting uncertainty in the first wave's predictive power based on fewer samples during this early time period.

## SARS-COV-2 sequencing and identification of variants of concern

Since COVID-19 waves have been largely driven by new variants of concern we wanted to understand the utility of sequence SARS-CoV-2 from wastewater. To better understand if ES can be used to identify VOCs circulating in river and a defunct WWTP, we carried out amplicon sequencing (see Methods) using the Nanopore MinION for 90 samples with the lowest diagnostic cycle threshold (Ct) between 30.2–38.8, which translated to viral genome copies/liter (gc/l) between 140735–1434 gc/l and produced 85 sequences. Although this is now commonly done in HIC with formal sewage systems it was unclear if we could identify VOCs from mixed environmental samples with low viral loads. Of our sequenced samples, only 20/86 samples had >50% genome coverage and 56/86 samples had >20% genome coverage at 20x depth (at least 20 reads mapped). To confirm our sequencing results and compare sequencing methodologies we carried out further matched sequencing of 68 samples using the EasySeq method (see Methods) and an Illumina MiSeq or NovaSeq (Fig S8). Of these

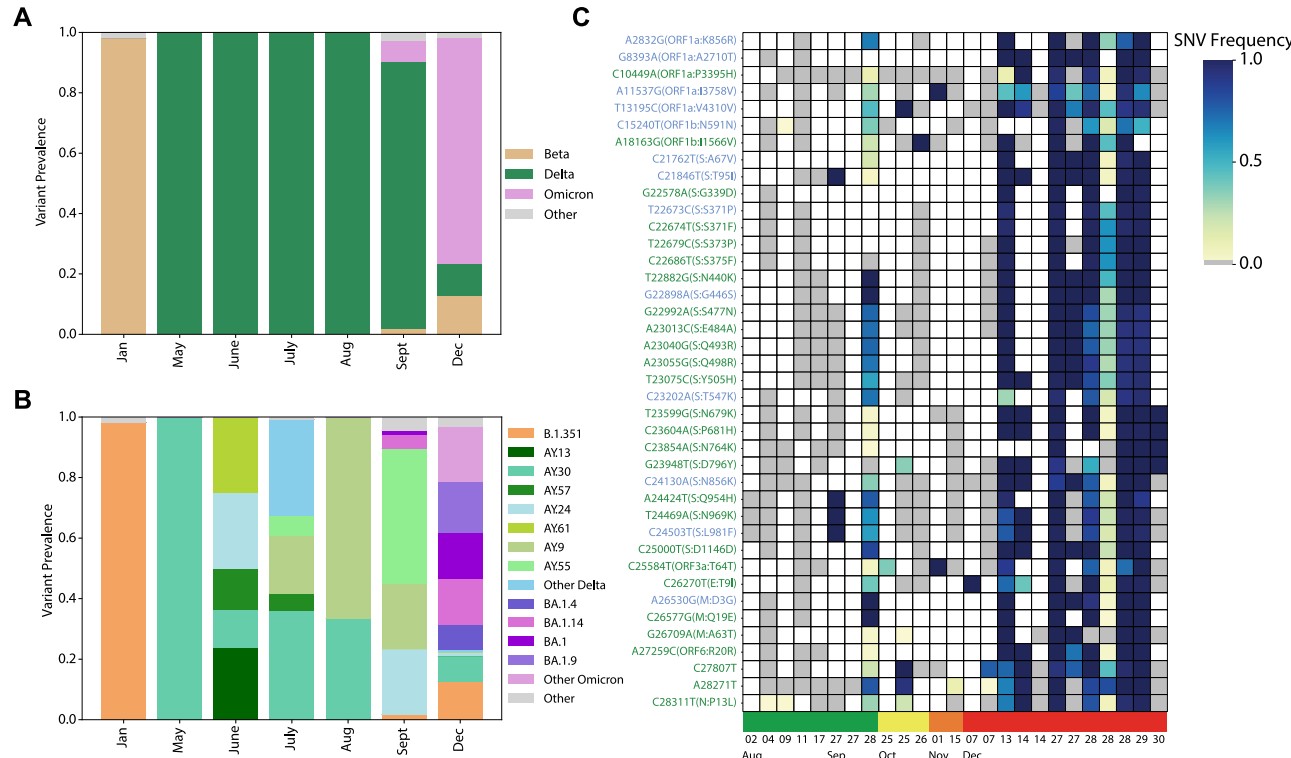

**Fig. 3 | SARS-CoV-2 variant in wastewater identified key VOCs before observed in the patient population. A** Summary of VOC detected by month using Freyja and **B** the VOC sub-lineage by month, and **C** Omicron SNPs show putative early detection of the VOC. The heatmap shows all BA.1 Omicron SNPs on the y-axis (blue=BA.1-specific, green=BA.1/BA.2 shared mutation) and individual ES samples overtime on the x-axis where months are by color. In September samples have some key Omicron SNPs but lack the full repertoire of SNPs which become dominant by December.

samples, 27/68 had >50% coverage and 42/68 samples had >20% coverage at 5x depth showing that low viral load greatly affected both sequencing methods.

We were able to sequence SARS-CoV-2 from both informal sewage and river water, important for ES in communities with limited to no refined sewage treatment centers. To identify circulating VOCs, we utilized the Freyja[25] analysis tool (Fig. 3A, B), which has been shown to effectively recover VOC frequencies from mixed wastewater samples with as low as 50% genome coverage. Freyja accounts for partial observation of mutational signatures, as expected due to factors including degradation in the environment and contaminants like PCR inhibitors[25,36], Freyja leverages partial observation of variant-specific mutational signatures (Fig. 3C, Fig S9) to estimate variant frequency at the single nucleotide polymorphism (SNP) level.

We first identified VOCs on the following dates: Beta (Jan 19th, 2021), Delta (May 18th, 2021) and Omicron (28th, Sept 2021) using both Nanopore MinION and Illumina data (Fig. 3A). Detection of Beta was consistent with clinical data; however, Delta was detected in wastewater a week before the first clinical detection in Blantyre (May 26th, 2021). Our initial analysis identified Omicron over two months before we observed a clinical case of Omicron in Malawi (Dec 6th, 2021), and 6 weeks before the first identification of this new variant globally (Nov 9th, 2021)[37,38] (Fig. 3B, C).

At face value the two genomes from September appeared to be Omicron by Freyja and one was identified as Omicron using the ARTIC to Pangolin computational pipeline[38]. Upon deeper analysis of the two putative Omicron genomes from wastewater samples collected in September 2021, we found that one majority consensus genome had multiple SNPs found in a clinical case (a BA.1.14 lineage virus) from Malawi collected in January 2022 and sequenced a few weeks before our wastewater sample was sequenced (Fig S10a). We re-sequenced the wastewater sample from the original water sample and the

re-sequence did not match the clinical sample and did not have the full Omicron repertoire of SNPs (Fig S10b), confirming our suspicion of contamination.

For the second putative Omicron positive sample from September 2021, we further investigated how to analyze a sample with two VOCs present. For this sample Omicron was estimated to be only about 10% of total virus present, we attempted to produce a unique genome by leveraging physically linked mutations specific to Omicron, SNP-frequency linkage (i.e., similar mutation frequency) and by filtering out reads corresponding to circulating Delta lineages, dominant VOC in the sample. Since genomes from wastewater are a mixture of multiple viruses, identifying a consensus genome requires consideration of additional factors including physically linkage of SNPs, SNP frequency, uniqueness of SNPs to known VOCs, and SNPs that are linked to multiple VOC. These considerations can greatly change the resulting interpretation and evolutionary dynamics over time. Using a Bayesian phylodynamic approach, we tested the effect of three different consensus calling approaches on the estimated sampling date of the virus (i.e., while blinding our model to the "known" sampling date of the virus) (Fig S11). If we included all physically linked and SNP frequency-linked mutations, we estimated sampling date in early 2022 (median Feb 5, 95% HPD = (Jan 9–Feb14)), in line with the global spread of Omicron and not indicative of early emergence in Malawi. When considering only physically linked mutations we found a median estimated sampling date of November 15th, 2021 (95% HPD = Oct 3, Jan 22) consistent with possible early Omicron circulation in Malawi prior to November 2021.

Freyja analysis uncovered some examples of cryptic transmission where Beta continues to pop up throughout the year and Omicron does not fully replace Delta until December (Fig. 3A) which was not identified in our patient data. Among samples with ≥20% genome coverage 15/51 contained more than one VOC and for samples with

≥50% genome coverage 5/28 samples contained more than one VOC. This timeline of cryptic detection and then the emergence of dominant VOCs is consistent with sequencing data from patients in Malawi where new VOCs become dominant in the patient population. Although we have low numbers and low genome coverage, we do see examples of cryptic transmission of known VOCs. We detect Beta intermittently until the end of 2021 and we continue to detect Delta even after Omicron moved through the population (Fig. 3A). Our current patient genomic surveillance failed to detect the continued circulation of these VOCs[37] but we detect cryptic Beta spread via wastewater. We do believe these to be genuine detections of cryptic Beta spread, we observe mutations across the genome with both physical and SNV frequency linkage including S: K417N, E484K, D215G, DEL 241/243, ORF3a:S171L, and ORF8:F120V, as well as a handful of additional mutations such as G28457A, C7392T, and C4276T, suggestive of significant evolution of the Beta variant beyond the main wave. We note that although these may seem to be surprisingly late detections of Beta, there are multiple high-quality records of Beta variant detections in nearby South Africa[39] around this time (e.g., EPI_ISL_10646387, EPI_ISL_7545672).

Finally, to understand how the ES sample genomes relate to patient data from Malawi as well as SARS-CoV-2 more globally, we constructed a maximum likelihood phylogenetic tree for all recovered Delta variant viruses. We used patient derived genomes from a time period similar to our samples with a bias toward genomes from Africa. For Delta the ES samples largely cluster with genomes from patients' samples from Malawi and samples from South Africa (Fig S12). This analysis highlights the utility of tracking VOC in wastewater as a powerful tool to detect known VOCs but due to the low viral load and low sample numbers we are unable to confidently detect emerging VOCs in the dataset.

## Discussion

We demonstrate that environmental surveillance of SARS-CoV-2 has the potential to be an important public health tool in a low-income setting where there is often no formalized sewage system and little clinical surveillance. We detected SARS-CoV-2 as soon as we started sampling in May 2020, but we never found the common viral control PMMoV highlighting this is not the best internal control in Malawi and likely other countries. We did find Hf183 was a good indicator of fecal contamination but in future studies we would expand our controls to include extraction controls like MS2 or an equivalent spike-in to minimize false negatives. We describe city-wide detection of SARS-CoV-2 in river systems, highlighting any source of water with fecal contamination can be used for viral detection. This demonstrates that communities without adequate sanitation systems, which is most of world, can sample rivers and semi-formal sewage systems to generate actionable information about SARS-CoV-2 trends. Our sample processing protocols, and analysis methods provide an easy-to-follow inexpensive workflow to perform wastewater-based pathogen surveillance in any setting.

We initially oversampled Blantyre, using multiple locations to capture a high percentage of the population and determine where virus was accumulating. Using hotspot, cluster analysis and percent positivity of sampling location we were able to identify 25 maximally informative sites that cover around 70% of the population that we will use for future studies, greatly reducing the overall cost of this surveillance. In an urban setting with no formal sewage system over sampling can help establish key areas with fecal contamination and therefore wastewater sites with detectable pathogens. As each urban area will be unique, we recommend oversampling for one year. By establishing the minimal number of informative sites, the inexpensive ES approach can run in parallel with sentinel community surveillance to provide an early warning system for public health officers to scale surveillance and public health interventions in response to environmental detection.

Rates of sample positivity correlated with clinical prevalence of COVID-19 in Blantyre, Malawi. Peaks of positivity in our ES samples preceded peaks in cases in the population when compared to both active and passive surveillance approaches except for during the Delta wave which showed similar trends between all datasets. This is consistent with trends observed in wealthier countries based on wastewater surveillance of refined sewage systems where early warning signals hover around 2–5 weeks[9,40–42]. Our modeling shows that for clinical prevalence above 5%, positivity in the ES data continue to trend upward and are predictive or at the very least mirror increases in the population. Utilizing this early warning signal community health centers and hospitals can scale supplies and staff to counteract the total burden on the healthcare system. In addition, public service announcements can be used to try and minimize close contacts, protect vulnerable people, and decreases surges in the population.

We demonstrate that genomic analysis from rivers and a defunct WWTP is possible in a setting with no refined sewage. By sequencing wastewater, we can obtain a snapshot of circulating VOCs in the population – a powerful and inexpensive tool for tracking outbreaks. Prior to the COVID-19 pandemic there had been no sequencing in Malawi, and this work helped establish genomic surveillance within the country. Whilst we were able to detect diverse VOCs from wastewater, our experience highlights the complexities of consensus calling for mixed samples, especially as new VOCs arise while older viral lineages are still circulating. This work established Beta, Delta, and Omicron VOCs circulating in the Southern region of Malawi, but our work is also a cautionary tale on the level of scrutiny needed to identify emerging VOCs. Despite significant efforts to separate processing of environment and clinical samples, pathogen sequencing of wastewater samples was not always performed in real-time, making it difficult to distinguish between actual early VOC detection and contamination. Our work does provide key analysis steps to disentangle mixed samples and create consensus genomes from VOCs with low genome coverage and depth. As many VOCs will originate in countries with low genomic surveillance, sequencing of wastewater has the potential to identify emerging SNPs and VOCs as noted by previous groups[25,43] but this requires a conservative approach. When VOCs are known, targeted sequencing can also be an inexpensive method to confirm physically linked SNPs. Early VOC detection in wastewater could be a powerful tool but sequencing needs to be carried out in real-time, both to enable timely public health interventions and to limit confounders for analyses, an issue that has plagued multiple SARS-CoV-2 genomic surveillance studies even from clinical samples.

ES has clear potential to act as an inexpensive[29] early warning system in the surveillance of SARS-CoV-2 and potentially other outbreaks in LICs with limited community surveillance but many factors need further consideration. The continuous shed of SARS-CoV-2 in Malawi may also be due, in part, to high levels of enteric diseases and immunocompromised populations[44]. This sustained shedding may also be driving the cryptic transmission of the Beta VOC we observe in wastewater but since we have no longitudinal shedding data from the HIV patient population this is only conjecture. ES is also identifying viral shed from the animal reservoir. Our work has shown that E. coli and K pneumoniae colonize throughout households in people, places and animals[45,46] and previous work by others has shown animals can acquire and pass SARS-CoV-2 to humans[47] therefore we are capturing the One-Health picture of SARS-CoV-2 infection.

Wastewater surveillance has the potential for identifying other environmentally dependent pathogens as we've shown through the detection of S. Typhi and there is potential to track multiple viral and bacterial pathogens longitudinally for little cost. We are currently evaluating the temporal trends of vaccine derived Polio and Rotavirus as well as other enteric viruses and bacteria at minimal extra cost. ES is a rapid and cost-effective tool to predict peaks in transmission and identify VOCs. Whilst future work must explore the impact of the water

conditions (pH, organic composition, flow rate), sampling strategies, concentration, sequencing methods, and environmental influences, ES is already an effective solution to tracking outbreaks and re-emergence of long-standing viral threats.

## Methods

### Ethics

Wastewater samples, although not human subjects, were collected under the ethical waiver P.07/20/3089 from the College of Medicine Research Support Centre (CoMREC). No identifiable information was used for the estimations of detection frequency between ES and the population. Nevertheless, active surveillance data was collected under CoMREC P.08/20/3099 and the Liverpool School of Tropical Medicine Research Ethics Committee (LSTMREC 21-058). Passive surveillance was collated by the District Health Office and no identifiable information was used for this analysis.

### SARS-CoV-2 detection methods technical development

We utilized our previously established ES program for *Salmonella enterica* serovar Typhi[31,32] to determine if we could detect SARS-CoV-2 in wastewater[48]. We first spike wastewater with $2 \times 10^8$ copies of SARS-CoV-2 genomic block (Table S1) and compared two widely used concentration methods, polyethylene glycol (PEG)[49] and skimmed milk flocculation (SMF)[49,50] recovery (Fig S13b). We recover about half of the viral particles and there was no difference between PEG and SMF. Although both methods are comparable based on our work we used a genomic fragment of SARS-CoV-2 for this technically work and not a full viral particle which is unavailable in Malawi. We next did a limit of detection utilizing a SARS-CoV-2 genomic block on 3 biological (separate wastewater samples) x 3 technical replicates and found we could detect samples with ≥50 genomic copies in a 30 mL sample. We then tested nine positive samples using unfiltered grab samples and filtered samples (effluent from a 0.45micron filter) and found the unfiltered samples had slightly lower PCR cycle thresholds / higher gc/l recovery (Fig S13a). This difference is not significant and further analysis of filtering samples to first remove bacteria should be more deeply considered but for our work we did not include the filtering step. Finally, we also squeezed Moore swabs places for S. Typhi, PEG concentrated and tested for SARS-CoV-2 but did not find any positive samples. Other passive collection methods like auto samplers are not possible in our environment as they require power or a battery and clog easily due to the level and size of debris in the water. We therefore chose grab samples due to cost and straightforwardness in the field and PEG due to more predictable availability of consumables and laboratory infrastructure. We did not include a viral spike-in as we were unable to ship live virus (ex. MS2) to Malawi due to very limited shipments and periods with no flights. Therefore, we may have false negatives in our dataset but since we consistency detected SARS-CoV-2 in line with clinical peaks be believe the false negatives are minimal.

### Sampling strategy

For 2020 (phase 1) we utilized seven collections sites based on sites with positive *S.* Typhi samples from our previous study[30]. In 2021 we scaled our collection to 112 sites spanning 22 areas of Blantyre, Malawi (Fig S1c, Supplementary Data 1). In brief site selection was undertaken using a GIS-based framework, all river confluence points within the city were selected. Geographical catchment areas were then generated using a topological dataset which was created using publicly available elevation data and the standardized AGREE watershed delineation approach. Population density was assigned to the identified catchment areas using WorldPop and high-resolution settlement layer datasets. Medium- and high- population density areas were selected, and other regions within the city that had industrial or agricultural land use, were removed. The city was then divided into 500 m polygons, with catchment areas mapped. If more than one GPS location was in the same rectangle, on the same river or water system, the location furthest upstream within the rectangle was selected as a candidate sampling location. These candidate sites were further stratified by population density and assigned to small, medium, or large categories. Priority sites were then decided based on which location had the greatest population served estimate per category.

These sites were packaged into a dataset including their GPS coordinates and topological information, and site identifier number, before on-site field assessment to ensure viability of the location before ES could begin. The site's proximity to the nearest vehicle access; the accessibility of the water by foot with sampling equipment; potential hazards; and the likely availability of sufficient water year-round were considered at this point. When a list of viable sampling locations was compiled, they were checked to ensure all areas of the city of interest were sufficiently covered, and where necessary, alternatives were provided where the original site was not viable for any of the reasons outlined above.

The only exception to this was the defunct Manase wastewater treatment plant (WWTP) where we sample the inlet, lagoon, and outlet to the Mudi River. Manase WWTP is one of the lowest points of the city (Fig S1a,b) and is the accumulation of multiple rivers therefore has high fecal contamination and accumulation of organic matter. Only two other collections sites (No ID 19 and Southwest 1) are downstream of the Manase WWTP (Supplementary Data 1).

Field teams visit around 40 sites/week plus Manase WWTP weekly. Grab samples are collected in 50 mL falcon tubes, to ensure a physical sample could be matched to its metadata, such as GPS location, date, sample type and downstream RT-PCR results, we utilized barcodes with dates and a unique identifier that were scanned into a sim-enabled tablet utilizing the KoBoToolBox database platform (https://kf.KoBoToolbox.org/)[51]. Samples from each location were collected weekly from May 14th - Dec 18th, 2020 (phase 1) with gaps in the collection largely due to safeguarding of the field team as the COVID-19 pandemic unfolded in Malawi. During Phase 2 we started collecting Jan 3rd, 2021, with full scale up by mid-February, 2021 of 112 sites collected about every 2 weeks or as needed due to changes in river dynamics and safety of the field team 40-50 sites/week (Fig S1c, Supplementary Data 1). If a site was deemed unsafe to collect either due to heavy rains, access, or personal safety they were either eliminated or missed until the site became safe again.

### Concentration of samples

We adopted polyethylene glycol (PEG) concentration methods based on work by Philo et al.[52]. Briefly, we added 3gram of PEG (VWR-UK APOSBIA2204-500G) and 0.68gram of NaCl (VWR-UK ACRO447302500) to a 30 mL grab sample in a 50 mL Falcon tube (Appendix 1). We manually shake by hand for 30 s or until the PEG and NaCl is dissolved. Up to 20 samples are then spun at 1200 g, 4 °C (1200 g – range from 800-2000 g) for 2 h. We did test decreasing the spin time to one hour and see little difference in SARS-CoV-2 detection. After centrifugation we discard supernatant trying to remove as much liquid as possible without dislodging the pellet and then add 200uL sterile 1x PBS (PH 7.4, Merck P4417-100TAB) to the pellet and vortex for 2 min. Using PBS, we make a final volume of 500ul. Samples were then transferred to a (locking) 1.5 Eppendorf tubes or cryovials and stored at −80 °C prior to RNA extractions. We find the viral particle is more stable in PEG than as RNA, so we recommend long term storage prior to RNA extraction.

### SARS-CoV-2 detection

PEG samples were extracted using either the Qiagen RNA viral extraction kit or the Qiasymphony-DSP mini kit 200 (Qiagen, UK) with offboard lysis. RNA was tested for SARS-CoV-2 using the CDC N1 assay (IDT) and qScript 1-step master mix (VWR-UK 733-2234) (10ul master mix, 1.5 primer/probe (FAM), 3.5ul water, and 5ul RNA) with a

positive and negative control. We also ran a standard curve using a serial dilution of a SARS-CoV-2 genomic fragment from 100,000 to 10 copies/ul. We utilized the starting volume of 30 ml, the average volume after concentration and standard curves with known genome copies to Ct based on a SARS-CoV-2 genomic fragment to calculate the genome copies of each sample per liter. We interpolated a sigmoidal curve based on three standard curves with known genome copies to calculate the genome copies based on the Ct of each positive sample (listed in Supplementary Data 2). This was dependent on the size of the final pellet. We add PBS to try and standardize the final volume of the pellet, but pellet size is largely dependent on total organic material in the original sample (i.e., dirtier water often led to larger pellets).

We also tested samples for Pepper Mild Mottle Virus (PMMoV) – a standard control in USA laboratories[33] but after testing multiple water and fecal samples and using 2 different primer/probe sets[42,53] (F:GAGTGGTTTGACCTTAACGTTTGA, R:TTGTCGGTTGCAATGCAAGT, and P:/5Cy5/CCTACCGAAGCAAATG/3IAbRQSp/ and F:GCTGAAGG TTGGTACTTGTA, R:TCAGGTCGGCTATGTATCAT, P:5Cy5/TGGATGAG CAGCGAACGGGTGA/3IAbRQSp/) we found zero positive sample. The same samples were positive for SARS-CoV-2 and Hf183 so there are both detected virus and bacteria in the samples. In Malawi there are very few varietals of peppers, and we believe PMMoV is not a virus present at high enough levels to be detected in wastewater. This is important for other studies in countries where plant viruses many be different. We also tested matched water samples collected from the same time and place as part of our S. Typhi work (manuscript in prep) for Hf183 utilizing the previously published[34] primers and probes: F:ATC ATGAGTTCACATGTCCG, R:CTTCCTCTCAGAACCCCTATCC, P:FAM/ CTAATGGAACGCATCCC/BHQ-1/).

## Statistical methods for SARS-CoV-2 from ES

We utilized 2 clinical datasets to compare signals of SARS-CoV-2 in wastewater. The datasets include an active surveillance cohort described in Chibwana 2023[54]. In brief, we recruited outpatients presenting to two large primary healthcare facilities in Blantyre from November 2020-March 2022. Patients with signs of COVID-19 as well as matched patients presenting with non-COVID-19 symptoms were recruited. Nasopharyngeal swabs were collected and testing utilizing the CDC-N1 assay identifying both symptomatic and asymptomatic individuals utilizing a barcodes LIMS system. The second passive dataset was COVID-19 positive numbers per day per health clinic captured by the District Health Office (DHO). The main limitation to this dataset is all cases were captured on a paper-based tracking sheets that were then collated in excel. We noticed specific dates (ex. right before Christmas holidays) when there was a larger amount of overall data on 1 or 2 days and then minimal data for weeks. The DHO did chase up discrepancies in the data with specific healthcare clinics but there was limited ability to track the accuracy of this database, therefore, we focused on the active dataset.

**Modeling ES detection rates.** We modeled the pattern of ES detection rate across all sites through the study period using a logistic generalized linear model, using a b-spline to capture temporal changes in detection rate. Binned Pearson residuals were generated by week to assess model fit, and knots were iteratively selected to minimize the binned weekly residuals, which were visually assessed by generating plots of weekly residuals over time.

**Statistical model.** There may be wave-specific differences in ES as it relates to clinical prevalence, due to changes in the sensitivity of the ES system, the relative amplitudes between waves (driven by healthcare seeking/ symptom severity differences) and delay between shedding and healthcare seeking (timing of symptoms), therefore we allow for wave-specific intercepts and lags in the model.

First, we estimated the lags between clinical prevalence and ES detection rates for each of the three waves. We utilized a quasi-binomial GLM to correct for overdispersion in the binomial GLM. We model the clinical prevalence at day $t$ as:

$$Y_t \sim Quasi\,Binomial\,(u_t)$$
$$\log\left(\frac{u_t}{1-u_t}\right) = \alpha + \beta^* ES_{t-lag} \tag{1}$$

Where $u_t$ is the clinical prevalence at time $t$, and $ES_{t-lag}$ represents the detection rates at the lagged time points, we investigated lags from −21 (3-week lead time) to 56 (8 weeks) to assess the best-fit lag for each wave. Finally, we include the lagged ES time series for each wave in a full model again utilizing a quasi-binomial GLM:

$$Y_t \sim Quasi\,Binomial\,(u_t)$$
$$\log\left(\frac{u_t}{1-u_t}\right) = \sum_{w=1}^{3} (\alpha_w + \beta_w^* ES_{t-lagw})^* Z_{w,t} \tag{2}$$

$Z_{w,t}$ is an indicator variable that is equal to 1 when t is contained in wave w, 0 otherwise and $lagw$ represents the estimated lag from the previous model.

We tested the robustness of these estimates to uncertainty in the ES time series. We generated 1000 realizations from the multivariate normal distribution parameterized by the ES regression model covariates. Next, we tested the significance of these realizations as predictors in the full model and summarized the significance of the covariate across this uncertainty.

## Comparing peaks between datasets

We aimed to summarize the differences between the two clinical dataset and the ES data, with regards to the timing of the maximum detection rates, or peaks. For each dataset, we fit a b-spline to each time series and extracted the timing of the peak for each wave. To account for uncertainty in the fitted model, we generated 1000 realizations from the multivariate normal distributions parameterized by these models (Fig S4) and extracted the timing of peaks for each realization. Using this data, we generated a 95% tolerance interval for the timing of each peak.

## Sequencing and analysis of SARS-COV-2

For positive samples Ct<39 (1150 gc/l), cDNA was generated using a 2-step process using Superscript IV (ThermoFischer) or a 1-step process and using LunaScript (NEB). For Nanopore MinION sequencing carried out in Malawi we utilized a modified ARTIC sequencing protocol V2. From May 2020 - December 2021 ARTIC V3 primers were used and in January 2022 we switched to V4 primers. Matched samples were also sequenced using an adopted EasySeq method (https://www. protocols.io/view/wastewater-sequencing-using-the-easyseq-rc-pcr- sar-81wgb7bx3vpk/v2 and https://www.nimagen.com/gfx/Covid19/ protcol-NimaGen-covid-wgs_v201.pdf) on an Illumina MisSeq or NovaSeq. When possible, we generated new cDNA using the EasySeq method which has a concentration step before cDNA generation. We used a 1.8x SPRI beads (Beckman Coulter). We found this yielded higher cDNA per ul. If we could not regenerate the cDNA using the EasySeq protocol, we utilized cDNA generated for the original Nanopore MinION sequencing. After cDNA generation the Easyseq method was carried out using either a V3 or V4 amplicon primer set.

For Nanopore MinION sequencing analysis we adapted the CLIMB/ ARTIC analysis pipeline to be carried out locally in Malawi[38]. Briefly, FAST5 data was processed using Guppy v5.0.7 including guppy_basecaller. Bam files were generated from FASTQs by assigning barcodes using guppy_barcoder, eliminating sequences lacking barcodes at both ends and using medaka ARTIC field bioinformatics pipeline. Available at: https://github.com/artic-network/fieldbioinformatics. Finally, consensus

genome (FASTA) files were generated using the original Wuhan genome based on the ARTIC pipeline[38].

For the VOC per sample analysis, we utilized the Freyja methods[25] which incorporates viral genetic diversity and infers relative abundance of lineage-defining mutations to deconvolute mixed samples. We used.bam files from both the Nanopore MinION and Illumina sequencing and the Freyja workflow (v1.3.10) and packages found at https://github.com/andersen-lab/Freyja.

We performed consensus calling only on samples with >50% genome coverage to determine VOC (as estimated by Freyja) and called SNVs with 50% or greater frequency. We then aligned consensus sequences to the Delta and Omicron VOC genomes separately and analyzed these with genomes from a similar time range from patient samples derived from Malawi, elsewhere in Africa, as well as samples from around the world. We aligned to reference using minimap2 (v2.24)[55] with gofasta (v1.1.0)[56] and used *masking* to remove homoplastic sites[57]. Maximum likelihood tree inference was generated using IQ-TREE2(v 2.2.0.3)[58] and tree rooting, and visualization were done using the toytree package. Heatmaps were generated utilizing known SNPs associated with the Omicron BA.1 and BA.2 lineages.

Bayesian date sampling analyses for putative early-Omicron samples was done using BEAST 1.10.4[59]. We used an HKY substitution model, a strict molecular clock, and an exponential growth coalescent tree prior. Analyses were performed on a background set of 332 early Omicron BA.1 sequences of high quality, plus the draft consensus sequence. Date sampling was performed using an uninformative uniform prior for the sample collection date. For date sampling of each candidate consensus sequence, we performed 200 million Markov Chain Monte-Carlo steps, with the first 20% of steps discarded as burn-in. Effective sample size was greater than 200 for all model parameters for all three cases.

### Reporting summary

Further information on research design is available in the Nature Portfolio Reporting Summary linked to this article.

## Data availability

The SARS-CoV-2 genomes generated in this study have been deposited in the NCBI Sequence Read Archive under BioProject ID PRJNA887942: PATH ES of SARS-CoV-2 -MLW. The raw including dates, GPS coordinates and PCR results are provided in Supplementary Data 2.

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

## Acknowledgements

This publication is based on research funded by PATH, the Bill & Melinda Gates Foundation (grant number 583722 and OPP1155752). K.G.B. is funded by a NIH Fogarty Fellowship (K01TW010853). Technical assistance from staff at The Centre for Genomic Research, University of Liverpool, UK Funding, Department of Health and Social Care, UK DHSC UK (2020_097) to (SP). The views expressed in this publication are those of the authors and not necessarily those of the Department of Health and Social Care. The findings and conclusions contained within are those of the authors and do not necessarily reflect positions or policies of the funders. We would like to acknowledge support and input from David Boyle, Sophie Magnet, Vajra Allan Mariana Sagalovsky and Supriya Kumar. We would also like to thank our technical staff at the Malawi-Liverpool Wellcome Trust.

## Author contributions

K.G.B., and N.F. conceived of the study with technical support from J.C.M. C.B.U. mapped all sites and performed the hotspot and cluster analysis. J.R. developed the kobo tracking system and identified collection sites. K.G.B, O.K., O.M., J.R., G.K., H.T., S.M., M.G.C., K.C.J., K.J., J.C., B.C., T.S.N. and N.F. established the wastewater dataset and/or the active and passive patient surveillance datasets. J.G and P.D. carried out the modeling analysis with help from C.C. who collated and cleaned all data. K.G.B, O.K., C.A., E.C., J.C., and S.P. carried out the sequencing. K.G.B., J.I.L, P.M.A., C.T-T., and K.G.A. carried out sequencing analysis. K.G.B, N.F, J.I.L, J.G, and K.G.A wrote the manuscript. All authors contributed to the editing of the manuscript.

## Competing interests

To our knowledge no author has a competing interest to this work including but not limited to financial and non-financial interest, paid or unpaid advocacy, patents, commercial employment related to this work.

## Additional information

[1]Malawi-Liverpool-Wellcome Clinical Research Programme, Kamuzu University of Health Sciences, Blantyre, Malawi. [2]Department of Immunology and Infectious Diseases, Harvard TH Chan School of Public Health, Boston, MA, USA. [3]Broad Institute of MIT and Harvard, Cambridge, MA, USA. [4]Department of Vector Biology and Tropical Disease Biology, Liverpool School of Tropical Medicine, Liverpool, UK. [5]Department of Immunology and Microbiology, The Scripps Research Institute, La Jolla, CA, USA. [6]Institute for Disease Modeling, Bill & Melinda Gates Foundation, Seattle, WA, USA. [7]Department of Clinical Sciences, Liverpool School of Tropical Medicine, Liverpool, UK. [8]Medical Research Council Centre for Global Infectious Disease Analysis, Department of Infectious Disease Epidemiology, School of Public Health, Imperial College London, London, UK. [9]Department of Organismic and Evolutionary Biology, Harvard University, Cambridge, MA, USA. [10]Blantyre District Health Office, Blantyre, Malawi. [11]Department of Clinical Infection, Microbiology and Immunology, Institute of Infection, Veterinary and Ecological Sciences, University of Liverpool, Liverpool, UK. [12]NIHR Health Protection Research Unit in Gastrointestinal Infections, University of Liverpool, Liverpool, UK. [13]Department of Environmental and Occupational Health Sciences, School of Public Health, University of Washington, Seattle, WA, USA. [14]CHICAS, Lancaster Medical School, Lancaster University, Lancaster, UK. [15]Public Health Institute of Malawi, Lilongwe, Malawi. [16]Scripps Research Translational Institute, La Jolla, CA, USA. [17]Department of Pathology, Kamuzu University of Health Sciences, Blantyre, Malawi. [18]School of Medicine, University of St Andrews, St Andrews, UK. ✉e-mail: Kayla.Barnes@lstmed.ac.uk

