## [Peer Review File · Nature Communications]

Utilizing river and wastewater as a SARS-CoV-2 surveillance tool in settings with limited formal sewage systemsReviewers' Comments:

Reviewer #1:

Remarks to the Author:

[Summary]

This manuscript highlights SARS-CoV-2 surveillance in sewage and river water in Malawi, a country with limited community-based COVID-19 testing capacity. The authors investigate the efficacy of using river and informal sewage wastewater to track SARS-CoV-2 spread in communities and predict variant of concern (VoC) emergence. The study spans May 2020 and January 2022, which overlaps the global spread of Delta (B.1.617.2 and AY.*) and Omicron (B.1.1.529 and BA.*) VoCs. Authors report that the environmental surveillance of the SARS-CoV-2 provides an earlier detection signal for VoCs and hence can help track the spread of the virus. The authors also investigate two early occurrences of Omicron in the environmental samples in Malawi and conclude that those are likely results of contamination.

Overall the paper presents a commendable example of the value of environmental surveillance of SARS-CoV-2 that goes beyond traditional wastewater surveillance and explores possibilities in resource-limited settings. The authors describe the capability they stood up and implemented and the immense benefits. Areas of improvement include additional discussion on experimental validation and limitations of computational tools included in their approach and better positioning within the context of previous work in non-wastewater treatment plant surveillance.

[Major comments]

1) Major contribution to the study's novelty is the use of river water instead of the standard sampling at the influent of a wastewater treatment plant. However, the study fails to reference related work that has attempted to use river water as one of the sources for the environmental surveillance of SARS-CoV-2, for example: <https://www.sciencedirect.com/science/article/pii/S0048969720339279>, <https://pubs.acs.org/doi/full/10.1021/acsestwater.2c00065>, <https://www.sciencedirect.com/science/article/pii/S0048969722009081>, and <https://iris.paho.org/handle/10665.2/54988> (<https://www.medrxiv.org/content/10.1101/2020.10.21.20215434v1>). To better place this study in the context of previous work, the authors should briefly discuss these previous studies, potential differences in approaches, and any other differentiating qualities of the current work.

2) The authors use Freyja as the main tool for detecting VoCs. While an excellent tool, the authors fail to mention any potential limitations of this approach. For example, a brief discussion of how well it's expected to perform amidst heavy sample degradation in the environmental setting, as supported by authors' observations on the breadth and depth of coverage, coverage gaps, and amidst recombinant lineages.

3) The authors mention that "Freyja analysis uncovered some examples of cryptic transmission where Beta continues to pop up throughout the year and Omicron does not fully replace Delta until December". Is it possible that the Beta detections are false positives by Freyja? A bit more evidence as to why the authors have confidence that this was a cryptic transmission event and not misclassification would strengthen this result.

4) With respect to sample numbers described in "SARS-COV-2 sequencing and identification of variants of concern." and available on the bioproject, some clarification is required. The authors state that 90 samples were run on the MinION, but "only 24/86 had >50% coverage". Then on the provided NCBI bioproject, only 20 SRAs are available. Please clarify.

[Minor comments]

a) Lines 150-153: what statistical test was used to obtain the p-value?

- b) Line 170: delta wave -> Delta wave
- c) Lines 200-202: What do the dates in parentheses represent? It needs to be explained in the legend.
- d) Line 206: Nanopore MinION, not Minion or minion (also in other places in the manuscript)
- e) Lines 209-211: Clarify that x% coverage means the percent of the genome covered by reads (i.e., breadth of coverage).
- f) Figure 3B is hard to read. Figure 3A can be improved by having a version in which Delta (AY.*) and Omicron (BA.*) sublineages are collapsed into a single color/label.
- g) Exact parameters and versions of all software tools should be made explicit.

Reviewer #2:

Remarks to the Author:

There is limited information on wastewater-based epidemiology in low-income countries to monitor the transmission of COVID-19. A few points of clarity are needed to understand this study and additional comments are made below -

There were 2 phases of environmental sample collection - phase 1 (limited environmental sampling, May 2020-Jan 2021) and phase 2 (expanded environmental sampling, Jan 2021-May 2022). While a range of river sites were selected, some were only sampled from once or twice. In comparison, the defunct WWTP was sampled from the most (n=233). It is unclear how far apart the various river sites are from each other and as such this sampling approach may influence the identified trends. Moreover, the results may be affected by seasonal trends (rainy vs dry season) over the long timespan. Grab sampling is preferred to composite sampling in low resource settings, however passive samplers are now well established as they are deployed in wastewater to sorb the virus over time. There is no mention of this possible approach. It is not clear which rivers were downstream of the defunct WWTP and how this was considered in the analysis.

Clinical case information was presented from two different data bases – one active and one passive. Line 182 states that the Blantyre District Health Office (DHO) was available from the beginning of environmental sampling until January 2022 while line 440 states it was available from beginning of environmental sampling until September 2021. Please advise. For the COVID-19 case prediction, it seems only the active database was used. In Table 2, when comparing peaks from ES and the clinical datasets, there is a single date given for the peak but it's not clear how the peak was defined and what the dates in brackets refer to. Lastly, most references appear in the introduction and there are limited references comparing similar findings, even if from different settings.

Reviewer #3:

Remarks to the Author:

(Note that I do not have the specific expertise to critique the genomic analysis. Please ensure another reviewer is comfortable doing this)

The manuscript presents data and analysis of SARS-CoV-2 in low resource setting, and the potential utility of ES. This is likely one of the first comprehensive analyses (although note this paper from SA <https://www.medrxiv.org/content/10.1101/2023.02.13.23285226v2.full.pdf+html> although the methods are quite different), illustrating what information can be gained from doing ES in low resource settings. Key things I note from the analysis:

1) SARS-COV-2 is detectable from ES, although at a low concentration (or high Ct) even though

little/no information is provided about the sites

2) ES positivity correlates with cases, even though little information about the quality of clinical cases is provided

3) variant information is provided, illustrating the potential for early warning of VOC

Largely, the data collection and analysis is conducted well and is innovative. However, greater detail and thought needs to be provided for an improved epidemiological perspective. The points above (especially 1 and 2) could be stronger if more information was provided and included in analysis of the ES data. Examples include details of site characteristics, and an investigation of factors that affect effect detection. While some investigation has been done it doesn't feel like the field (or the project?) can use it to improve activities or analyses going forward? Why is this? Or, has this just not been explained very well? It is not clear whether the further analysis has not been considered, or the data were not collected? I especially find it difficult to understand why ct values are provided rather than gene concentration - especially as one figure in the suppl has this - the reader has to do this conversion every time so why not help the reader?

I'm providing these comments in the hope that they are constructive - they are quite large in number but I do think the ms would really improve by addressing them.

The manuscript could be greatly improved by framing the analysis around epidemiological investigation and making use of epidemiological theory and robust statistical investigation. Currently the analysis is difficult to follow - the narrative is not clear. I would suggest framing it around 1) phase I and identifying what lab methods and data collection is informative, 2) phase II where *only a subset of data* are considered further. If the 'uninformative' sites are still used, it is important to explain why...eg. Perhaps for some settings this is the only option?

Specific issues that must be addressed in future submissions:

Details required for reporting of qPCR data are not provided.

- Limit of detection and limit of quantification
- Were repeat samples taken / replicate samples in lab?
- It is unclear why the data were not translated to gene copied per litre?
- qPCR details (refer to MIQE guidelines)

The distinction between river and sewage samples is difficult to understand from what is presented. Even though the results are not described well, it does seem like an important element of the analysis. It is not clear what the randomness (or not) is of sewage samples and how this might influence results? If at the initial stage it was identified that sewage samples are more likely to be positive, why are the results always presented as a combined value?

- The spatial analysis sounds quite interesting but it is difficult to view the figures (1C-F) because they are small and I do not know the geography of Malawi (and most readers will not either).
- It is not clear what data are used in the spatial analysis - the full data or the "highly-informative" locations?
- The authors refer to site reduction - was this actually done - I do not see this in the data presented? To what extent was the "highly informative" sites the sewage samples instead of river samples? I would recommend having a figure illustrating the locations of the highly informative and not informative sites. It suggests that going forward the collection will change...specifying the details of this would be useful because then it is more clear what has been learnt from this initial set of data collection and analysis.

Further specific comments:

- L143 "Validaton"
- I think this needs to be re-written. It is initially presented as comparison between 1 ES dataset, and

1 clinical dataset. But actually there are 2 datasets (which are poorly described) and really, the previous section suggested that ES exists either as sewage or river..

- Please fully describe the models you use, including what data you have and what parameters there are..? This is attempted in the appendix but is difficult to follow. For example, what is Y_t ? Do you assume it is independent?

- The results are not described particularly well. Figure S2 is difficult to interpret.

- Introduction - The aim of the analysis is not really specified. It is merely said that the 'utility' is described. I recommend stating what the aim(s) of the analysis is.

Results/Methods

- This would benefit by having an overview at the start describing the order of analysis that you are going to present. There are lots of 'first', 'further', 'finally' using these loses the narrative rather than improving because there are so many.

- The ES sites are described as "rivers, informal sewage systems and a defunct wastewater treatment plant", but when I look at Table S1 (which is very useful) I see that all except 1 site are described as "river". It seems like those labelled 'river' need to be more clearly described as 'river' or 'informal sewage'. Is this the only distinction of sites?

- Phase 1 details refer to no sig difference, but no test details are provided. Only a figure is provided. Are the samples paired?

- The tests for media use is the only example where the data have been converted to gene copied per litre. Why do this conversion just here? What was the test?

- Was there a rationale behind ES expansion? It was the expansion based on an epidemiological question, or a pragmatic one (more resources available)?

- L125. So all of phase 2 was using grab samples, and PEG? (Pls clarify)

- Were any additional wastewater metrics collected, eg. Coliform counts / pH / etc?

Figure 1.

- Should add in arrows illustrating phase 1 and 2 in plots A and B.

- For all figures involving ES the authors need to include a secondary x-axis, presumably for % positive. Without this, the information is confusing.

Grammar/spelling

L335: 'with' instead of 'will'?

L638: 'over time' instead of overtime

Reviewer #4:

Remarks to the Author:

The paper fills an important gap that is how informal sewage or rivers can be used for surveillance of SARS-CoV-2. This is of great importance in low-income countries or areas with no access to wastewater treatment. Additionally, many of these countries have limited clinical surveillance increasing the importance of environmental surveillance. This paper contributes to our understanding of how SARS-CoV-2's evolution can be monitored using environmental waters contaminated with wastewater.

Concerns

- The authors state that active and passive detection lags behind ES detection except during the Delta wave. However the lack of comprehensive clinical surveillance can be the real reason for this delay.

- The choice to use grab samples and not composite samples can have an effect on the consistency of the data. It can also impact detection/non detection depending on the sample extraction timing (Augusto MR et al.,. Sampling strategies for wastewater surveillance: Evaluating the variability of SARS-CoV-2 RNA concentration in composite and grab samples. *J Environ Chem Eng.* 2022 Jun;10(3):107478. doi: 10.1016/j.jece.2022.107478

- Samples should be spiked before extracting RNA to validate the method used and the capacity of the SARS-CoV-2 virus to survive in informal sewage or environmental waters contaminated with sewage (Farkas K et al. (2023) Wastewater-based monitoring of SARS-CoV-2 at UK airports and its potential role in international public health surveillance. *PLOS Global Public Health* 3(1): e0001346. <https://doi.org/10.1371/journal.pgph.0001346>)

- The authors should mention a powerful strategy when sequencing samples that contain contributions from many individuals. Targeted sequencing has shown to be of great value to identify circulating VOC. Targeted sequencing also has the potential to identify sequences in environmental/wastewater that have not been detected in clinical samples (Smyth, D.S.,et al. Tracking cryptic SARS-CoV-2 lineages detected in NYC wastewater. *Nat Commun* 13, 635 (2022). <https://doi.org/10.1038/s41467-022-28246-3>)

- The possible contribution of certain hosts (immunocompromised patients for example) that shed SARS-CoV-2 for long periods should be mentioned (Leung WF, et al. COVID-19 in an immunocompromised host: persistent shedding of viable SARS-CoV-2 and emergence of multiple mutations: a case report. *Int J Infect Dis.* 2022 Jan;114:178-182. doi: 10.1016/j.ijid.2021.10.045. Epub 2021 Oct 29. PMID: 34757008; PMCID: PMC8553657)

- The possibility that animals could be contributing to the SARS-CoV-2 viruses present in environmental waters should be discussed (Rao SS, et al. Susceptibility of SARS Coronavirus-2 infection in domestic and wild animals: a systematic review. *3 Biotech.* 2023 Jan;13(1):5. doi: 10.1007/s13205-022-03416-8. Epub 2022 Dec 11. PMID: 36514483; PMCID: PMC9741861

- The detailed explanation of the likely contamination of a couple of ES samples does not add much value to the paper.

Overall, the paper is an important contribution to the field. The evidence that SARS-CoV-2 can be detected in informal sewage and rivers contaminated with sewage is clear and compelling. These findings should support efforts to implement ES in low income countries. However, the concerns listed above need to be addressed before the paper is accepted for publication.

Please find our response in Blue under each comment

Reviewer #1 (Remarks to the Author):

[Summary]

This manuscript highlights SARS-CoV-2 surveillance in sewage and river water in Malawi, a country with limited community-based COVID-19 testing capacity. The authors investigate the efficacy of using river and informal sewage wastewater to track SARS-CoV-2 spread in communities and predict variant of concern (VoC) emergence. The study spans May 2020 and January 2022, which overlaps the global spread of Delta (B.1.617.2 and AY.) and Omicron (B.1.1.529 and BA.*) VoCs. Authors report that the environmental surveillance of the SARS-CoV-2 provides an earlier detection signal for VoCs and hence can help track the spread of the virus. The authors also investigate two early occurrences of Omicron in the environmental samples in Malawi and conclude that those are likely results of contamination.*

Overall the paper presents a commendable example of the value of environmental surveillance of SARS-CoV-2 that goes beyond traditional wastewater surveillance and explores possibilities in resource-limited settings. The authors describe the capability they stood up and implemented and the immense benefits.

Areas of improvement include additional discussion on experimental validation and limitations of computational tools included in their approach and better positioning within the context of previous work in non-wastewater treatment plant surveillance.

[Major comments]

1) Major contribution to the study's novelty is the use of river water instead of the standard sampling at the influent of a wastewater treatment plant. However, the study fails to reference related work that has attempted to use river water as one of the sources for the environmental surveillance of SARS-CoV-2, for example:

<https://www.sciencedirect.com/science/article/pii/S0048969720339279>,

<https://pubs.acs.org/doi/full/10.1021/acsestwater.2c00065>,

<https://www.sciencedirect.com/science/article/pii/S0048969722009081>, and

<https://iris.paho.org/handle/10665.2/54988>

(<https://www.medrxiv.org/content/10.1101/2020.10.21.20215434v1>). To better place this study in the context of previous work, the authors should briefly discuss these previous studies, potential differences in approaches, and any other differentiating qualities of the current work.

We have now reworked the introduction and included these publications to better place our work within the global context. We thank the reviewer for flagging these publications and preprints.

2) The authors use Freyja as the main tool for detecting VoCs. While an excellent tool, the authors fail to mention any potential limitations of this approach. For example, a brief discussion of how well it's expected to perform amidst heavy sample degradation in the environmental setting, as supported by authors' observations on the breadth and depth of coverage, coverage gaps, and amidst recombinant lineages.

We thank the reviewers for raising this point. While sample degradation and contaminants like PCR inhibitors impact sequencing quality and do play a significant role in Freyja's ability to resolve lineage frequencies from environmental sample mixtures, we and others have shown

that Freyja produces accurate estimates for samples with 50-60% genomic coverage. However, and results can vary depending on where in the genome we have sufficient sequencing depth. For our work since we used both minion and Illumina platforms (long and short read sequencing) we are confident our VOC outputs are real. We note this was not clear in the manuscript and we have clarified this in the text (Line 335-340):

“We were able to sequence SARS-CoV-2 from both informal sewage and river water, important for ES in communities with limited to no refined sewage treatment centers. To identify circulating VOCs, we utilized the Freyja analysis tool²⁵ (Fig 3A-B), which has been shown to effectively recover VOC frequencies from wastewater for samples with as low as 50% genome coverage. Freyja accounts for partial observation of mutational signatures, as expected due to factors including degradation in the environment and contaminants like PCR inhibitors.”

3) The authors mention that “Freyja analysis uncovered some examples of cryptic transmission where Beta continues to pop up throughout the year and Omicron does not fully replace Delta until December”. Is it possible that the Beta detections are false positives by Freyja? A bit more evidence as to why the authors have confidence that this was a cryptic transmission event and not misclassification would strengthen this result.

This an excellent question, as contamination could also generate false “cryptic” variant detection events. We have added text to more clearly explain our findings (line 424-432) To examine the Beta detections from December in greater detail, we looked at reads containing Beta specific mutations in those samples. We observe as suite of Beta-specific “mutations across the genome with both physical and SNV frequency linkage including S: K417N, E484K, D215G, DEL 241/243, ORF3a:S171L, and ORF8:F120V, as well as a handful of additional mutations such as G28457A, C7392T, and C4276T, suggestive of significant evolution of the Beta variant beyond the main wave. We note that although these may seem to be surprisingly late detections of Beta, there are multiple high quality records of Beta variant detections in nearby South Africa around this time (e.g., EPI_ISL_10646387, EPI_ISL_7545672).”

4) *With respect to sample numbers described in "SARS-COV-2 sequencing and identification of variants of concern." and available on the bioproject, some clarification is required. The authors state that 90 samples were run on the MinION, but "only 24/86 had >50% coverage". Then on the provided NCBI bioproject, only 20 SRAs are available. Please clarify.*

After reviewing our data we only have 20 samples that meet the >50% coverage. We had two samples close to 50% and two repeat samples, hence the confusion on our end. We have changed this in the text.

[Minor comments]

a) *Lines 150-153: what statistical test was used to obtain the p-value?*

We have added this information in the text (line 248). A Wald test was used to obtain the p-value.

b) *Line 170: delta wave -> Delta wave*

This has been corrected

c) Lines 200-202: What do the dates in parentheses represent? It needs to be explained in the legend.

The dates in parentheses represent the 95% tolerance interval. This has been added in the legend and table and description was added in the methods.

d) Line 206: Nanopore MinION, not Minion or minion (also in other places in the manuscript)

Thank you for this flag. We have corrected this in all locations

e) Lines 209-211: Clarify that x% coverage means the percent of the genome covered by reads (i.e., breadth of coverage).

We have now clarified in the text (line 326-327) that % coverage here is defined as % 20x coverage, the percentage of the genomic sites with at least 20 reads mapped.

f) Figure 3B is hard to read. Figure 3A can be improved by having a version in which Delta (AY.*) and Omicron (BA.*) sub lineages are collapsed into a single color/label.

We thank the reviewer for these suggestions and have now reworked Fig 1 and Fig 3 moving the positivity by area (Fig 1) and larger heatmap and BEAST analysis (Fig 3) to supplementary where they can be a full page. Now, Fig 3A shows the deconvolution at the VOC level, while Fig 3B provides additional information on VOC sublineages (with low prevalence lineages grouped to further streamline the figure). To make the old Figure 3B easier to read, we have restricted the new iteration, now Figure 3C, such that we only show BA.1 mutations frequencies over the Delta-to-Omicron transition, starting in August 2021. We have moved our BEAST analysis figure to S11 and we now provide the complete heatmap in supplement in Figure S9.

We feel this makes the figures more clear without losing important results in the main text.

g) Exact parameters and versions of all software tools should be made explicit.

These have now been added throughout the methods

Reviewer #2 (Remarks to the Author):

There is limited information on wastewater-based epidemiology in low-income countries to monitor the transmission of COVID-19. A few points of clarity are needed to understand this study and additional comments are made below -

There were 2 phases of environmental sample collection - phase 1 (limited environmental sampling, May 2020-Jan 2021) and phase 2 (expanded environmental sampling, Jan 2021-May 2022). While a range of river sites were selected, some were only sampled once or twice. In comparison, the defunct WWTP was sampled from the most (n=233).

We have provided more information on how sites were chosen and then sampled during this study in the methods. We have also reworked the results to provide data from each phase and cumulatively to clarify.

It is unclear how far apart the various river sites are from each other and as such this sampling approach may influence the identified trends.

We have expanded our methodology to include descriptions of the sampling approach that is also described in full in (Uzzell et al., 2021) and sampling strategy. This includes more information about the geography of the city about why some sites were under sampled and other sites were over sampled. We have also added supplementary Figure S2 to show site location in relation to the elevation of the city which is a key factor in our approach. Blantyre is surrounded by three small mountains so there are large elevation differences between the collection points.

Moreover, the results may be affected by seasonal trends (rainy vs dry season) over the long timespan.

Yes – this is an important point for many LIC that experience a pronounced rainy season including Malawi. We have added text and a supplementary Fig S1d that combines rain water data with our SARS-CoV-2 detection. Since our detection closely corresponds to clinical data and does not directly correspond to the rainy season we believe that seasonal trends have limited effect on the SARS-CoV-2 outbreak but we agree for most enteric pathogens - we expect temporal trends based on rainfall.

Grab sampling is preferred to composite sampling in low resource settings, however passive samplers are now well established as they are deployed in wastewater to absorb the virus over time. There is no mention of this possible approach.

We have expanded the methods to clarify why we chose grab samples and PEG. We have also included more information about using passive samplers including Moore swabs and autosamplers. There are limitations to these methods. For bacteria detection we find Moore swabs are the preferred choice but without the culture option we did not recover as viral particles in our pilot work. We now have an auto sampler in place to catch runoff from the hospital which requires a battery and full locked building so the sampler and battery are not removed but auto samplers are expensive, require maintenance and security as they are run off a battery - making this a less cost effective option for LICs.

It is not clear which rivers were downstream of the defunct WWTP and how this was considered in the analysis.

We have added more detail about our sampling strategy and why we had a high sampling for the WWTP. In brief, this site is one of the lowest points of the city and 3 major river systems end and accumulate at this point. The only sites downstream of the plant are on the Mudi river and are sites No ID19 and Southwest 1. We have added this in methods. We have also added a topographical map Fig S1a to better visually show how Blantyre is surrounded by 3 main peaks.

Clinical case information was presented from two different data bases – one active and one passive. Line 182 states that the Blantyre District Health Office (DHO) was available from the beginning of environmental sampling until January 2022 while line 440 states it was available from beginning of environmental sampling until September 2021. Please advise. For the COVID-19 case prediction, it seems only the active database was used.

Line 440 was a typo and we have reworded this sentence to clarify we had DHO data through January 2022. We did not receive further DHO data as they stopped tracking SARS-CoV-2 cases at the city level and many clinical sites stopped submitting results. This was largely due to

low numbers of SARS-CoV-2 as well as the DHO redirecting attention to a Polio vaccine campaign.

In Table 2, when comparing peaks from ES and the clinical datasets, there is a single date given for the peak but it's not clear how the peak was defined and what the dates in brackets refer to.

First we have reworked this section and now Table 2 is Table 1. We have clarified the methods beginning at line 441 and the header of Table 1 to define the peaks and the tolerance interval.

Lastly, most references appear in the introduction and there are limited references comparing similar findings, even if from different settings.

We have added additional reference throughout. Thank you for bringing this to our attention.

Reviewer #3 (Remarks to the Author):

(Note that I do not have the specific expertise to critique the genomic analysis. Please ensure another reviewer is comfortable doing this)

The manuscript presents data and analysis of SARS-CoV-2 in low resource setting, and the potential utility of ES. This is likely one of the first comprehensive analyses (although note this paper from SA <https://www.medrxiv.org/content/10.1101/2023.02.13.23285226v2.full.pdf+html> although the methods are quite different), illustrating what information can be gained from doing ES in low resource settings.

Yes, we work closely with Mukhlid and his team. The major difference is SA is an upper-middle income country and their work represents sampling from a functioning sewage systems while Malawi has no sewage system and is a low-income country.

Key things I note from the analysis:

1) SARS-COV-2 is detective from ES, although at a low concentration (or high Ct) even though little/no information is provided about the sites

Information about the sites has been expanded in the results, methods and discussion and we have added a map of sites showing elevation (Fig S1a). We have also included a database (Table S2) with information on each collection with genome copies per liter. See also response to reviewer 2 above.

2) ES positivity correlates with cases, even though little information about the quality of clinical cases is provided

We have expanded the methods to give more details about the clinical dataset we used in this work and reorganized our modeling analysis section to make it more clear how we compared to an active clinical dataset and passive clinical dataset. In the active surveillance cohort, we collected data from outpatients presenting at two primary healthcare facilities in Blantyre, Malawi, from November 2020 to March 2022. Eligible participants were aged >1month old, with signs suggestive of COVID-19, and those not suspected of COVID-19, from whom we collected nasopharyngeal swabs for SARS-CoV-2 PCR testing (Chibwana 2023). We are therefore

confident about the quality of case detection here. Further limitations of DHO data are described in methods and results. We include the DHO data because, although we observe issues with dates and number, this passive surveillance is very typical in LICs and even with its limitations corresponds to our ES patterns.

3) variant information is provided, illustrating the potential for early warning of VOC

Largely, the data collection and analysis is conducted well and is innovative. However, greater detail and thought needs to be provided for an improved epidemiological perspective. The points above (especially 1 and 2) could be stronger if more information was provided and included in analysis of the ES data.

Examples include details of site characteristics, and an investigation of factors that affect effect detection.

We have greatly expanded the details of the site selection, geography of Blantyre, rainfall data and sampling strategy in all sections including new supplementary figures and note this was flagged but other reviewers as well. This is not an epidemiological analysis. Although we do compare our data to two clinical surveillance programmes what we have analyzed are trends and how ES is useful in our setting.

While some investigation has been done it doesn't feel like the field (or the project?) can use it to improve activities or analyses going forward? Why is this? Or, has this just not been explained very well?

We find this comment surprising and unclear as this is one of the first comprehensive evaluations of ES of virus in an LIC. As we remark in the abstract, our work highlights wastewater can be used for detecting emerging waves, identifying variants of concern and function as an early warning system in settings with no formal sewage systems. The work generated considerable excitement within the Public Health Institute of Malawi, the main surveillance arm of the Ministry of Health, as evidenced by the presence amongst the authors of Dr Ben Chilima the outgoing director of PHIM and we have a direct collaboration with PHIM to establish ES in Lilongwe (the capital and largest city in Malawi) that largely stemmed from this work. We have also shared our protocol and experiences widely and ES is now being adopted by groups in Nigeria and Sierra Leone.

We have added additional information about next steps and to better inform future studies in Blantyre and beyond in the discussion including important things to consider when sampling in communities with no formal sewage system.

It is not clear whether the further analysis has not been considered, or the data were not collected? I especially find it difficult to understand why ct values are provided rather than gene concentration - especially as one figure in the suppl has this - the reader has to do this conversion every time so why not help the reader?

We have provided all data in both Ct and gc/liter in a new Table S2 that includes our full dataset. For our tech development we have edited Fig S13 so results are in gc/liter. We note: Ct is a widely used metric in many diagnostic assays including multiple publications of SARS-CoV-2 from patient samples.

We seek clarity on the reviewers question.... 'It is not clear whether the further analysis has not been considered, or the data were not collected'? What specifically is the reviewer referring to?

I'm providing these comments in the hope that they are constructive - they are quite large in number but I do think the ms would really improve by addressing them.

The manuscript could be greatly improved by framing the analysis around epidemiological investigation and making use of epidemiological theory and robust statistical investigation. Currently the analysis is difficult to follow - the narrative is not clear. I would suggest framing it around 1) phase I and identifying what lab methods and data collection is informative, 2) phase II where *only a subset of data* are considered further. If the 'uninformative' sites are still used, it is important to explain why...eg. Perhaps for some settings this is the only option?

We have clarified the two phases of collection and the active vs passive clinical dataset comparisons with significant edits the results and methods. For context in this response Phase 1 proof of principle collection was merely expanded to a larger Phase 2. We analyzed the full dataset by RT-PCR, in our modeling and for sequencing. We have also rearranged how we present our modeling data to better clarify the difference between the active and passive clinical surveillance datasets we utilized. We have also rearranged the figures and added additional figures to better frame Blantyre in the context of geography and rainfall.

We respectfully disagree this paper is not framed around epidemiological investigation. This study is based on a carefully considered set of sites for environmental surveillance (Uzzell et al), a well-constructed programme of active community surveillance (Chibwana et al) and by comparison to real-world data collected by the Blantyre DHO, Three different but recognized epidemiological approaches. It should also be noted that with the ES, we are not studying a disease condition, nor trying to establish its incidence or prevalence, but looking for the presence and trends of a pathogen in the environment. We have developed a bespoke environmental public health surveillance system which permits the early identification of SARS-CoV-2 in a setting where that capacity does not exist in the community and which clearly has the potential to generate actionable information for public health authorities in Malawi.

Regarding the statistical analysis, we agree that epidemiological investigation using environmental surveillance is the ultimate use-case for this type of system and data collection. However, due to locations such as Blantyre having no formal sewage systems requiring sampling of river systems, the utility of ES data for epidemiological investigation in this type of setting has to first be validated. Therefore, the primary goal of this statistical analysis was to validate the ES system as a reflection of the prevalence of disease as assessed through clinical surveillance. Now that ES as a measurement of disease burden has been validated for this setting, epidemiological investigation would be a useful next step.

Regarding the third point, all data has been used in the analysis, not a subset. As the reviewer flagged, this is a necessary process of site selection for environmental surveillance in systems with informal sewage and limited resources. We have expanded this in our sampling methods as the number one reason some sites are under sampled was safety of our team. This was largely related to changing waterways and accessibility of roads.

Specific issues that must be addressed in future submissions:

Details required for reporting of qPCR data are not provided.

- Limit of detection and limit of quantification

We have added these details in our methods in a new section 'limit of detection' followed by our other technical development PEG vs MFL, the use of PPMOV and Hf183

- Were repeat samples taken / replicate samples in lab?

The full dataset is based on one sample/timepoint/location.

For Sequencing data key samples with early omicron were extracted from a second water sample to ensure reproducibility of unique or important SNPs

- It is unclear why the data were not translated to gene copied per litre?

We have now provided a full Table (S2) with all of our results including gc/l

- qPCR details (refer to MIQE guidelines)

We have updated the methods section to describe the experimental design included all primer and probe sequences, cycling conditions, master mix recipes, primers/probes gblcok (Table S3) throughout the methods and through linked publications.

The distinction between river and sewage samples is difficult to understand from what is presented. Even though the results are not described well, it does seem like an important element of the analysis. It is not clear what the randomness (or not) is of sewage samples and how this might influence results?

Yes, we agree we were not clear. We have now used either river or defunct WWTP (sewage).

Only the Manase defunct WWTP was "sewage" but even that was largely from river run off. All other sites are technically "river" but in a country with zero piped sewage system and no functioning WWTP all river in urban areas likely has some human waste. This distinction is difficult to make so for the purpose of this manuscript we have clarified what rivers look like in Blantyre but kept the river designation because they are technically rivers.

If at the initial stage it was identified that sewage samples are more likely to be positive, why are the results always presented as a combined value?

We have clarified this throughout the manuscript and added more context for why the Manase defunct WWTP has so many positive samples and why we over sampled this site. For other sites these are all technically 'river' but we tried to highlight collection sites with known sewage contamination. All rivers in Blantyre will have some level of sewage contamination. In Tables and Figure S1c (formally Figure 1) the results separated by sample location and sample type and in the manuscript we highlight the % positivity from the defunct WWTP vs river. We have also added more information about controls we used including PMMoV and Hf183.

- The spatial analysis sounds quite interesting but it is difficult to view the figures (1C-F) because they are small and I do not know the geography of Malawi (and most readers will not either).

Thank you for flagging this. We agree and we have reworked Fig 1 and 3 to make these more clear. We have reconfigured Fig1 to provide more space (and readability) for the maps by moving a larger panel to supplementary (Fig S1c). We have added a topographical map, highly informative sites and rainfall data to Fig S1. We hope this makes the figures clearer.

- It is not clear what data are used in the spatial analysis - the full data or the “highly-informative” locations?

The full dataset May 2020 - May 2022 was used for the spatial analysis. We have clarified this in the figure legend and text. We have added Fig S1b to show the highly informative sites.

- The authors refer to site reduction - was this actually done - I do not see this in the data presented?

We have clarified - for this study there was no site reduction but the hotspot analysis gives us a better idea for future programmes. This has been clarified in the discussion.

To what extent was the “highly informative” sites the sewage samples instead of river samples? I would recommend having a figure illustrating the locations of the highly informative and not informative sites. It suggests that going forward the collection will change...specifying the details of this would be useful because then it is more clear what has been learnt from this initial set of data collection and analysis.

We have added a new supplementary Fig S1b as well as added this information throughout the text. To Note - these sites did not inform the sampling strategy for this work but did inform our collection going forward. We have expanded our insights based on this in the discussion.

Further specific comments:

- L143 “Validaton”

- I think this needs to be re-written. It is initially presented as comparison between 1 ES dataset, and 1 clinical dataset. But actually there are 2 datasets (which are poorly described) and really, the previous section suggested that ES exists either as sewage or river.

Thank you for this feedback and we agree as written this was not clear. We have rearranged this into two sections: to first highlight the differences between ES vs Active dataset and ES vs passive dataset with more description of the clinical surveillance and limitations. We then dive into the deeper modeling we did to home in on the question around ES as an early warning system based on the active dataset which we felt was more accurate to true positivity per day. We hope this makes this section clearer.

- Please fully describe the models you use, including what data you have and what parameters there are..? This is attempted in the appendix but is difficult to follow. For example, what is Y_t ? Do you assume it is independent?

We have expanded the methods section to include further detail about the data and models, and added new section headers to clarify. The ‘appendix’ supplementary figures are referred to in the results and methods sections, which we have edited for clarity. **We are not clear what the reviewer is referring to as Y_t ?**

- The results are not described particularly well. Figure S2 is difficult to interpret.

We have added more information in the legend to Fig S2 which is now figure S5. Further, we have added additional detail in the paragraph of the results describing Figure S5 (line 257).

- Introduction - The aim of the analysis is not really specified. It is merely said that the 'utility' is described. I recommend stating what the aim(s) of the analysis is.

Thank you for flagging this. We have now added a sentence to the last paragraph of the intro to describe our three main aims.

Results/Methods

- This would benefit by having a overview at the start describing the order of analysis that you are going to present. There are lots of 'first', 'further', 'finally' using these loses the narrative rather than improving because there are so many.

We Agree and we have reworked the Results and Methods and added more detail about site selection and each analysis.

- The ES sites are described as "rivers, informal sewage systems and a defunct wastewater treatment plant", but when I look at Table S1 (which is very useful) I see that all except 1 site are described as "river". It seems like like those labelled 'river' need to be more clearly described as 'river' or 'informal sewage'. Is this the only distinction of sites?

We agree with the reviewer this is confusing and we have clarified in the results and methods and added more detail to the table. We have removed the term 'informal sewage system' as this is the same as the defunct WWTP. In short all the rivers have some level of sewage but how much sewage runoff and where it occurs is very difficult to determine (and changes daily and seasonally).

- Phase 1 details refer to no sig difference, but no test details are provided. Only a figure is provided Are the samples paired?

We agree that this is confusing and we have reworked the methods section as this refers to the technical development we did to choose our sampling and detection strategy (Fig S13) which we have now greatly expanded in the results and methods adding the section (line 523) '**SARS-CoV-2 detection methods technical development**'

- The tests for media use is the only example where the data have been converted to gene copied per litre. Why do this conversion just here? What was the test?

We have now added gc/l in a section supplementary Table S2. For our technical development we spiked samples with exact gc/liter and we have reworked fig S13a to be gc/liter to be consistent.

- Was there a rationale behind ES expansion? le was the expansion based on an epidemiological question, or a pragmatic one (more resources available)?

The expansion happened for two main reasons

1. We wanted to cover close to 100% of the population

2. We wanted to better evaluate what water sources had SARS-CoV-2 and higher fecal contamination.

Phase two was also supported with awarded funding after we demonstrated we could detect SARS-CoV-2 in a limited number of sites

- L125. So all of phase 2 was using grab samples, and PEG? (Pls clarify)

First, we agree this was confusing and we have reworked the manuscript to clarify this.

To answer your question: Correct - we did initial tech development and found Grab (unfiltered) and PEG gave us the highest viral recovery for cost / availability of reagents. We used the same sampling strategy though the full collection (both Phase 1 and 2).

We have rearranged the Phase 1 and 2, reworked our figures and added to the sections to the methods to clarify tech dev, sampling strategy and our overall approach. We hope the significant changes in the results and methods make this clearer

- Were any additional wastewater metrics collected, eg. Coliform counts / pH / etc?

Not for this study but we have added this for future studies. At the time of this study we had no working meters and it took over 18 months to get one into the country.

Figure 1.

- Should add in arrows illustrating phase 1 and 2 in plots A and B.

We have added a colored bar to show the different phases of the study. Thank you for the suggestion.

- For all figures involved ES the authors need to include a secondary x-axis, presumably for % positive. Without this, the information is confusing.

The figures all show individual samples colored for either positive or negative RT-PCR (Fig 1A) so we have added a right-hand y-axis to show number of samples. For Fig 2 and Fig S3 only positive samples (which we have clarified in the legend) are graphed. Because total sample vary per day/week/month we cannot add a general % positivity as a second error bar as the denominator changes.

Grammar/spelling

Thank you for picking up these grammar mistakes:

L335: 'with' instead of 'will'?
corrected

L638: 'over time' instead of overtime
corrected

Reviewer #4 (Remarks to the Author):

The paper fills an important gap that is how informal sewage or rivers can be used for surveillance of SARS-CoV-2. This is of great importance in low-income countries or areas with no access to wastewater treatment. Additionally, many of these countries have limited clinical surveillance increasing the importance of environmental surveillance. This paper contributes to our understanding of how SARS-CoV-2's evolution can be monitored using environmental waters contaminated with wastewater.

Concerns

- The authors state that active and passive detection lags behind ES detection except during the Delta wave. However the lack of comprehensive clinical surveillance can be the real reason for this delay.

We completely agree that the limited surveillance and other health factors including delayed health seeking in Malawi, lack of robust infrastructure, clinical stuff, access to diagnostics and the overall young age of the population are playing major rolls in a delay in picking up new peaks of SARS-CoV-2. That said, numerous papers from high income countries have found ES gives a slight early warning signal even with the best clinical surveillance. We believe this is actually due to early shedding by asymptomatic carriers and symptomatic carriers that are not so unwell that they seek treatment.

In a low income setting with minimal community surveillance delays and limited surveillance will always be present, making the case that environmental surveillance in this setting plays a significant role in pathogen surveillance. We have tried to bring this out more in the discussion and have added citations to show our work is in line with HIC.

- The choice to use grab samples and not composite samples can have an effect on the consistency of the data. It can also impact detection/non detection depending on the sample extraction timing (Augusto MR et al.,. Sampling strategies for wastewater surveillance: Evaluating the variability of SARS-COV-2 RNA concentration in composite and grab samples. J Environ Chem Eng. 2022 Jun;10(3):107478. doi: 10.1016/j.jece.2022.107478

We did evaluate Moore swaps for SARS-CoV-2 detection which we have now described in the methods. We have found Moore swabs provide better detection for S. Typhi but we were never able to recover any SARS-CoV-2 More advanced composite samples using an autosampler is not possible in this setting. They require a battery to run and in our case that would require guards at all 100 locations plus the cost of equipment maintenance far exceeds anything a LIC public health agency could likely support. We also found the autosampler clogged very easily in our setting due to the large amount of particulate matter and garbage in our water system. The manuscript flagged is largely based on work from the Netherlands which is hard to extrapolate to our setting.

- Samples should be spiked before extracting RNA to validate the method used and the capacity of the SARS-CoV-2 virus to survive in informal sewage or environmental waters contaminated with sewage (Farkas K et al. (2023) Wastewater-based monitoring of SARS-CoV-2 at UK airports and its potential role in international public health surveillance. PLOS Global Public Health 3(1): e0001346. <https://doi.org/10.1371/journal.pgph.0001346>)

Thank you for this point and we have added more information about this to the discussion and methods. For context there were large time periods where no flights (except Embassy flights)

were leaving or entering Malawi. In addition, shipping live virus requires a cold chain that was very difficult to access during the Pandemic. Malawi was continuously put on and off the “red list” minimizing flights and therefore the movement of supplies. For this work we were unable to get MS2 or other spike-ins to Malawi that required dry ice for delivery, in our working going forward we are adding MS2. I have added this in the limitations and lessons learned in the discussion. This publication is from 2023 and was not available when we began this work in 2020.

The lack of MS2 (or any spike-in) means we may have some false negatives not false positives. Since we see consistent trends of positivity that correlate with clinical trends there is likely minimal false negatives in our dataset. MS2 on its own does not ensure all samples with SARS-CoV-2 will be detected. Therefore this does not change our overall findings. We feel we have addressed other key points and are presenting a well characterized and valid dataset.

Instead of spiking we tried to use the gold standard, PPMOV detection as an indicator of fecal contamination but after numerous RT-qPCRs attempts including 2 primer sets and testing the water samples and human stool samples we never found PPMOV. In Malawi there are only a few pepper varieties and spicy food is uncommon. Although this control works well in the USA we found it was not appropriate for this setting. We have added this in the methods because this will be an important note for other non-US countries. We did find Hf183 in samples both with SARS-CoV-2 and no SARS-CoV-2.

- The authors should mention a powerful strategy when sequencing samples that contain contributions from many individuals. Targeted sequencing has shown to be of great value to identify circulating VOC. Targeted sequencing also has the potential to identify sequences in environmental/wastewater that have not been detected in clinical samples (Smyth, D.S., et al. Tracking cryptic SARS-CoV-2 lineages detected in NYC wastewater. Nat Commun 13, 635 (2022). <https://doi.org/10.1038/s41467-022-28246-3>)

This is well noted and we have included this citation and expanded the discussion.

- The possible contribution of certain hosts (immunocompromised patients for example) that shed SARS-CoV-2 for long periods should be mentioned (Leung WF, et al. COVID-19 in an immunocompromised host: persistent shedding of viable SARS-CoV-2 and emergence of multiple mutations: a case report. Int J Infect Dis. 2022 Jan;114:178-182. doi: 10.1016/j.ijid.2021.10.045. Epub 2021 Oct 29. PMID: 34757008; PMCID: PMC8553657)

Yes – we discuss this often and at length. I have added the potential for higher shedding due to the immunocompromised population (which remains high >10% in Malawi) including adding the noted citation. This may also be the driver of the cryptic transmission of beta we see much later in Malawi than observed in HIC.

We also believe the high burden of enteric pathogens is also leading to higher viral load in wastewater.

We have expanded the discussion to highlight this need for further research (line 501-509)

- The possibility that animals could be contributing to the SARS-CoV-2 viruses present in environmental waters should be discussed (Rao SS, et al. Susceptibility of SARS Coronavirus-2

infection in domestic and wild animals: a systematic review. 3 Biotech. 2023 Jan;13(1):5. doi: 10.1007/s13205-022-03416-8. Epub 2022 Dec 11. PMID: 36514483; PMCID: PMC9741861

We agree and our work in the bacteria space (now cited and added to the discussion) has show how households including animals are colonized. We have added discussion that the ES is also picking up the animal reservoir and have included the above citation.

- The detailed explanation of the likely contamination of a couple of ES samples does not add much value to the paper.

Thank you for this comment and we have now reworked the results section. There has been a lot of scrutiny of SARS-CoV-2 sequences including recent retractions of major papers, so we are conscience that this level of detail is important for some readers, and we did not want to overstate our findings. We have now moved some redundant information to the Fig S10 legend to simplify this section.

Overall, the paper is an important contribution to the field. The evidence that SARS-CoV-2 can be detected in informal sewage and rivers contaminated with sewage is clear and compelling. These findings should support efforts to implement ES in low income countries. However, the concerns listed above need to be addressed before the paper is accepted for publication.

We thank the reviewers for their support. We feel based on reviewers feedback this manuscript is a much stronger and clearer manuscript.

Reviewers' Comments:

Reviewer #1:

Remarks to the Author:

The authors have adequately and comprehensively addressed all of my feedback and I thus have nothing further to add.

Reviewer #2:

None

Reviewer #3:

Remarks to the Author:

The authors have done an great job of responding to my comments, and imo the other authors. The manuscript is now much clearer and easier to understand, strengthening the merit of the research. My comments about the epidemiological merit of the study and analysis was concerning how the study was described (ie. communication). The scientific approach is much better explained.

The comments below are just my views and recommendations to the authors, I do not require further correspondence on these,

- Reporting Ct versus gc/L (or equivalent). While the authors are correct that both are reported and so perhaps they could continue to report Ct, I do not recommend this. It is best practice to report gc/L s that samples across the same study but different batches/labs/etc are comparable, and also across studies. In a PATH commissioned "Equals" review it is recommended to report gc/L (but note, I not sure how public this study is, but presumably some of the authors have been involved.)

- "What is Y_t ?"... here I am referring to the equations under "predictive model"...apologies, equation writing does not happen in the reviewer platform; Y_t refers to 'latex' shorthand for writing equations, ie. Y subscript t. I do still think some more definitions would help the reader understand the model (and is usual practice for stats analysis), ie. Define what Y_t , u_t , ES_t and parameters are, even if it seems obvious to you, it might not be to others. Some of this is present in table 2, but actually further descriptions (eg. Units of measurement) would help with interpretation. Otherwise, things get lost in translation.

Reviewer #4:

Remarks to the Author:

Overall, the paper is an important contribution to the field. The evidence that SARS-CoV-2 can be detected in informal sewage and rivers contaminated with sewage is clear and compelling. These findings should support efforts to implement ES in low income countries. The authors have revised the manuscript responding to many of the reviewers comments. However, some important issues still need to be addressed.

Regarding methods:

1- The methods used to choose between PEG and SMF is based on SARS-CoV2 genomic block. This is not comparable to viral particles that need to be concentrated from wastewater. The value of FigureS13B is questionable. It is totally understandable that the lack of resources forced some difficult choices but this data is of very little value. In the same figure the gc/l values on the y-axis seem to be too low to be detected.

2- Based on the unique characteristics of wastewater the analysis must include physically linked SNPs. The authors should not elaborate on potential early appearance of the Omicron VOC in Malawi. We suggest to remove the "early detection of Omicron"from the discussion.

3- Target sequencing should be mentioned as an approach to get more depth in physically linked SNPs specially when sequencing potentially degraded wastewater samples.

Minor issues

1- Freezing samples at -80C as describe in Appendix 1 has been shown to lower the titers of SARS-CoV2.

2- In line 127 Figure 1A is described as having geographical information. It does not.

3- In lines 141-142 the authors can't conclude that there is no difference between filtered and unfiltered samples based on the data shown.

4- Table S2 should report either only genomic copies/liter or both Cts and genomic copies/liter.

5- Line 335: Across the paper they replaced informal sewage with defunct WWTP but in this line they mention informal sewage.

6- Line 432: The authors should include references for high quality records of Beta variant detection in South Africa

Response to REVIEWERS' COMMENTS

Reviewer #1 (Remarks to the Author):

The authors have adequately and comprehensively addressed all of my feedback and I thus have nothing further to add.

Thank you for taking the time to review this reworked manuscript

Reviewer #3 (Remarks to the Author):

The authors have done an great job of responding to my comments, and imo the other authors. The manuscript is now much clearer and easier to understand, strengthening the merit of the research. My comments about the epidemiological merit of the study and analysis was concerning how the study was described (ie. communication). The scientific approach is much better explained.

The comments below are just my views and recommendations to the authors, I do not require further correspondence on these,

- Reporting Ct versus gc/L (or equivalent). While the authors are correct that both are reported and so perhaps they could continue to report Ct, I do not recommend this. It is best practice to report gc/L s that samples across the same study but different batches/labs/etc are comparable, and also across studies. In a PATH commissioned “Equals” review it is recommended to report gc/L (but note, I not sure how public this study is, but presumably some of the authors have been involved.)

Thank you for this comment – we have now added gc/l when we discuss specific Ct(s) expect in the case of Figure S2 where we have added the gc/l range in the figure legend. In Figure S2, plotting gc/l gives an inaccurate visual of sample-to-sample variation as all our ES samples have low viral loads compared to what you would observe in clinical samples.

On this note we are part of a larger Africa and LIC consortium trying to better standardize ES work where autosamplers and other more sophisticated options are not possible. We very much agree standardization across platforms is needed.

- “What is Y_t ?”... here I am referring to the equations under “predictive model”...apologies, equation writing does not happen in the reviewer platform; Y_t refers to ‘latex’ shorthand for writing equations, ie. Y subscript t. I do still think some more definitions would help the reader understand the model (and is usual practice for stats analysis), ie. Define what Y_t , u_t , ES_t and parameters are, even if it seems obvious to you, it might not be to others. Some of this is present in table 2, but actually further descriptions (eg. Units of measurement) would help with interpretation. Otherwise, things get lost in translation.

Thank you for the clarity. We have re-organized the model methods section and added further detail on parameters including units of measurement. We feel this has added more information about our models for future interpretation.

Reviewer #4 (Remarks to the Author):

Overall, the paper is an important contribution to the field. The evidence that SARS-CoV-2 can be detected in informal sewage and rivers contaminated with sewage is clear and compelling. These findings should support efforts to implement ES in low income countries. The authors have revised the manuscript responding to many of the reviewers comments. However, some important issues still need to be addressed.

Regarding methods:

1- The methods used to choose between PEG and SMF is based on SARS-CoV2 genomic block. This is not comparable to viral particles that need to be concentrated from wastewater. The value of FigureS13B is questionable. It is totally understandable that the lack of resources forced some difficult choices but this data is of very little value. In the same figure the gc/l values on the y-axis seem to be too low to be detected.

We agree this early tech dev does not provide new data but merely shows either method is an option in an LIC setting. We have expanded the figure legend to clarify this by adding PEG shows “*modest but insignificant*” and “*but overall both methods are comparable in our setting*”.

In our methods we have also added the following sentence since as you rightly pointed out we did not use a full viral genome for this work starting on line 429.

“Although both methods are comparable based on our work we used a genomic fragment of SARS-CoV-2 for this technically work and not a full viral particle which is likely unavailable in Malawi”

We have chosen to leave the figure in supplementary as we feel this information both in text and figure is important as we receive this question from collaborators often.

2- Based on the unique characteristics of wastewater the analysis must include physically linked SNPs. The authors should not elaborate on potential early appearance of the Omicron VOC in Malawi. We suggest to remove the “early detection of Omicron” from the discussion.

Apologies – but we have not added the statement “early detection of Omicron” in the discussion or results. On the contrary we have added extensive discussion highlighting the caveats of sequencing and detection of VOCs. Is it possible this was in track changes from the previous version?

3- Target sequencing should be mentioned as an approach to get more depth in physically linked SNPs specially when sequencing potentially degraded wastewater samples.

We have now added the sentence in the discussion line 383-285. “When VOCs are known, targeted sequencing can also be an inexpensive method to confirm physically linked SNPs.”

For our work the minion amplicon sequencing is a form of targeting sequencing and if we identify “physically linked” SNPs this is based on the SNPs being on the same amplicon or fragment (for Illumina) and not across different amplicons or fragments, but we agree with minion there is more room for sequencing error and targeted sequencing is a great inexpensive tool and does confirm physical linkage.

Minor issues

1- Freezing samples at -80C as describe in Appendix 1 has been shown to lower the titers of SARS-CoV2.

Yes – we agree the freeze thaw can decrease the overall viral Ct. We have now added a note in the appendix as real-time processing is always preferred.

- a. ***Note: storing at -80°C can decrease the detectable viral particles so whenever possible extract and proceed to PCR on the same day.***

2- In line 127 Figure 1A is described as having geographical information. It does not.

This does not appear in the clean version of the manuscript and we have checked that Fig 1A and Fig S1a are correctly described in the manuscript

3- In lines 141-142 the authors can't conclude that there is no difference between filtered and unfiltered samples based on the data shown.

We have added the sentence in line 436-437 “*This difference is not significant and further analysis of filtering samples to first remove bacteria should be more deeply considered but for our work we did not include the filtering step.*”

In the clean version of the manuscript filter vs unfiltered does not appear until the methods and is no longer in line 141-142

4- Table S2 should report either only genomic copies/liter or both Cts and genomic copies/liter.

In the version submitted Table S2 reports both gc/l (column F) and Ct (column E).

5- Line 335: Across the paper they replaced informal sewage with defunct WWTP but in this line they mention informal sewage.

This was removed in the clean version and likely appeared in a confusing way in the track changes – apologies as there were considerable track changes. We note this has been corrected across the manuscript.

6- Line 432: The authors should include references for high quality records of Beta variant detection in South Africa

We have now added the Tegally (Nature 2021 et al paper) now citation 39 in the manuscript.